# Distributed inference for high-dimensional convoluted rank regression

Xinyu Zhang

School of Mathematical Sciences, Capital Normal University, Beijing, 100048, China

Xu Guo

School of Statistics, Beijing Normal University, Beijing, 100875, China

Bingye Yang [*]

School of Mathematical Sciences, Peking University, Beijing, 100871, China

## Abstract

Convoluted rank regression is recently developed as a powerful tool to deal with outliers and heavy-tailed noise data. However, there is still a lack of suitable methods for convoluted rank regression in a distributed system, which is becoming very common nowadays. To solve this issue, we investigate the distributed inference of the convoluted rank regression for high-dimensional data. We introduce two debiased estimators which are communication-efficient and computation-efficient, respectively. For these two introduced estimators, we derive the asymptotic representations and establish the Gaussian approximation results. To further reduce $\ell_1$ and $\ell_2$ errors, we also introduce sparse debiased estimators. Finally, simulation studies and real data analysis are conducted to illustrate our proposed methods.

**Keywords:** Debiased estimator, distributed data, high-dimensional convoluted rank regression

**Mathematics Subject Classification (2020):** 62J05

## 1 Introduction

With the rapid advancement of information technology, modern datasets are often characterized not only by high dimensionality but also by sheer size. Conventional centralized methods, where all datasets are stored and processed in a central computing facility, are often impractical in applications. Privacy or security constraints also make it challenging to process the entire dataset on a single central machine. Consequently, distributed computing systems become increasingly popular. Large-scale high-dimensional datasets are often characterized by outliers and heavy-tailed noise, posing new computational and storage challenges to conventional statistical methods. We are dedicated to advancing robust statistical inference procedures for high-dimensional distributed datasets.

---

[*]Corresponding author: byyang25@stu.pku.edu.cn

There are many developments to deal with outliers and heavy-tailed noise. Actually quantile regression and Huber regression are two typical choices for robustness considerations. Recent developments about these two regression models can be found in Belloni and Chernozhukov (2011), Wang et al. (2012), Fan et al. (2017), and Sun et al. (2020). Besides, rank regression recently receives a lot of attention due to its both robustness and high efficiency. Wang et al. (2020) made a deep theoretical investigation of high-dimensional sparse penalized rank regression. The non-smooth nature of the rank regression loss function limits the rank regression's popularity. Notably, Zhou et al. (2024) adopted the convoluted smoothing technique (Fernandes et al., 2021) and introduced a novel convoluted rank regression (CRR). They further proposed corresponding penalized estimators and derived their estimation error bounds. Since the penalized convoluted rank regression estimators in Zhou et al. (2024) cannot be directly used to make inference, Cai et al. (2025) further introduced a debiased estimator and developed inference procedures. However, when the sample size is very large, the computation of high-dimensional convoluted rank regression is still a big problem.

Directly aggregating all data encounters practical obstacles, including storage limitations, communication overhead, and privacy concerns. To address these issues, one-shot methods are commonly employed. For these methods, estimators computed on local machines are transmitted to a central site and aggregated to form a global estimator, as in Zhang et al. (2013), Rosenblatt and Nadler (2016) and Fan et al. (2019). The most popular and direct aggregation method is simple averaging, which averages the estimators obtained on each local machine, as in Zhang et al. (2013). Besides averaging, some other aggregating methods have been considered, such as meta estimators considering the variance of each local estimator developed in Liu et al. (2015) and Kullback-Leibler divergence method in Liu and Ihler (2014). Although one-shot approaches have the lowest communication costs, they suffer from several disadvantages, as discussed by Zhou et al. (2023). Alternatively there are some communication-iterative methods allowing a reasonable number of iterations, as in Shamir et al. (2014), Jordan et al. (2019) and Fan et al. (2023). Shamir et al. (2014) introduced a distributed approximate Newton-type method, which avoids transferring Hessian matrices across sites. Jordan et al. (2019) developed a communication-efficient surrogate likelihood framework for distributed data, where the key idea is to update the Hessian matrix on a single site only. For other important methods, kindly see the review by Gao et al. (2022) and Zhou et al. (2023).

In this paper, we investigate distributed inference methods for the high-dimensional convoluted rank regression. Motivated by the debiased techniques (see, for instance, Zhang and Zhang (2014); Van de Geer et al. (2014); Javanmard and Montanari (2014)), we develop debiased procedures for distributed convoluted rank regression. We make the following contributions.

- Firstly, we propose an averaging debiased estimator, which is one-shot approach and thus communication-efficient. The procedure involves obtaining debiased convoluted rank regression estimators on each machine and then averaging them.

- Secondly, we introduce a one-step debiased estimator, which is computation-efficient and requires only an estimator of the Hessian matrix on a single machine.

- Thirdly, we develop sparse averaging debiased estimator and sparse one-step debiased esti-

mator, which significantly reduce the $\ell_1$ and $\ell_2$ errors of the averaging debiased estimator and the one-step debiased estimator.

- Lastly, we establish theoretical results for our introduced estimators. For the averaging debiased estimator and the one-step debiased estimator, we derive their asymptotic representation, Gaussian approximation, and $\ell_\infty$ error bound. Meanwhile, for the sparse debiased estimators, we obtain their $\ell_1$ and $\ell_2$ error bounds.

The remainder of the paper is organized as follows. In Section 2, we develop statistical inference procedures based on the averaging debiased estimator. In Section 3, we develop statistical inference procedures based on the one-step debiased estimator. Section 4 reports numerical results from Monte Carlo studies. A real data example is presented in Section 5. Conclusions and discussions are provided in Section 6. All proofs and additional simulation studies are given in the Supplementary Material.

**Notations:** For a vector $\boldsymbol{x} = (x_1, \ldots, x_p)^\top$, let $\|\boldsymbol{x}\|_r = (|x_1|^r + \cdots + |x_p|^r)^{1/r}$ for $r \geq 1$ and $\|\boldsymbol{x}\|_\infty = \max_{1 \leq i \leq p} |x_i|$. For a matrix $\boldsymbol{A} = (a_{ij})_{n \times n}$, define $\|\boldsymbol{A}\|_r = \sup_{\|\boldsymbol{x}\|_r = 1} \|\boldsymbol{A}\boldsymbol{x}\|_r$, $\|\boldsymbol{A}\|_{\max} = \max_{1 \leq i,j \leq n} |a_{ij}|$ and $\|\boldsymbol{A}\|_\infty = \max_{1 \leq i \leq n} \sum_{j=1}^n |a_{ij}|$. $\lambda_{\min}(\boldsymbol{B})$ and $\lambda_{\max}(\boldsymbol{B})$ are the smallest and largest eigenvalues of a positive semidefinite matrix $\boldsymbol{B}$. The sub-Gaussian norm of a sub-Gaussian random variable $X$ is defined as $\|X\|_{\psi_2} = \inf\{t > 0 : \mathbb{E}[\exp(X^2/t^2)] \leq 2\}$. The sub-exponential norm of a sub-exponential random variable $Y$ is defined as $\|Y\|_{\psi_1} = \inf\{t > 0 : \mathbb{E}(\exp(|Y|/t)) \leq 2\}$. A $p$-dimensional random vector $\boldsymbol{X}$ is sub-Gaussian if and only if $\sup_{\|\boldsymbol{u}\|_2 = 1} \|\boldsymbol{u}^\top \boldsymbol{X}\|_{\psi_2} < \infty$. Let $\|\boldsymbol{X}\|_{\psi_2} = \sup_{\|\boldsymbol{u}\|_2 = 1} \|\boldsymbol{u}^\top \boldsymbol{X}\|_{\psi_2}$ be the sub-Gaussian norm of $\boldsymbol{X}$. For two numerical sequences $\{x_n\}$ and $\{y_n\}$, $x_n = O(y_n)$ if there exists $c > 0$ such that $|x_n| \leq c|y_n|$, and $x_n = o(y_n)$ if $\lim_{n \to \infty} x_n/y_n = 0$. For a sequence of random variables $\{X_n\}$, $X_n = O_p(1)$ if $\lim_{t \to \infty} \limsup_{n \to \infty} \mathbb{P}(|X_n| \geq t) = 0$, and $X_n = o_p(1)$ if $X_n$ converges to 0 in probability.

## 2 Inference based on averaging debiased estimator

Consider the following linear regression model:

$$Y_i = \boldsymbol{X}_i^\top \boldsymbol{\beta}^* + \epsilon_i,$$

where $\{Y_i\}_{i=1}^n$ are independent and identically distributed (i.i.d.) responses, $\{\epsilon_i\}_{i=1}^n$ are random errors, and $\{\boldsymbol{X}_i\}_{i=1}^n$ are i.i.d. $p$-dimensional random vectors with $\mathbb{E}(\boldsymbol{X}_i) = \boldsymbol{0}$ and $\mathrm{Var}(\boldsymbol{X}_i) = \boldsymbol{\Sigma} > 0$. The random vector $\boldsymbol{X}_i$ is independent of $\epsilon_i$. The unknown parameter of interest is $\boldsymbol{\beta}^* \in \mathbb{R}^p$. Let $\mathcal{S} = \{j : \beta_j^* \neq 0\}$ and $s = |\mathcal{S}|$.

Hettmansperger and McKean (2010) considered the following rank regression estimator for $\boldsymbol{\beta}^*$:

$$\arg \min_{\boldsymbol{\beta} \in \mathbb{R}^p} \frac{1}{n(n-1)} \sum_{i \neq j}^n |(Y_i - \boldsymbol{X}_i^\top \boldsymbol{\beta}) - (Y_j - \boldsymbol{X}_j^\top \boldsymbol{\beta})|.$$

They noted that the rank regression estimator performs well when the error distribution is heavy-tailed. However, the non-smooth nature of the rank loss function poses a significant computational challenge. Zhou et al. (2024) proposed CRR, which reduces the computational

burden of the rank regression estimator:

$$\arg\min_{\beta \in \mathbb{R}^p} \frac{1}{n(n-1)} \sum_{i \neq j}^n L_h\big((Y_i - Y_j) - (\boldsymbol{X}_i - \boldsymbol{X}_j)^\top \boldsymbol{\beta}\big),$$

where $L_h(u) = \int_{-\infty}^{+\infty} |u - v| K_h(v)\, dv$ with $K_h(v) = h^{-1} K(v/h)$. Here, $K(v)$ is a kernel function, and $h$ is a bandwidth. A smooth kernel function makes the convoluted rank loss function smooth. Define

$$\boldsymbol{\beta}_h^* = \arg\min_{\beta \in \mathbb{R}^p} \mathbb{E}[L_h\big((Y_i - Y_j) - (\boldsymbol{X}_i - \boldsymbol{X}_j)^\top \boldsymbol{\beta}\big)].$$

Now consider a distributed setting with $M$ local machines. Assume that the dataset, with sample size $N$, is equally distributed across the $M$ machines. Let $n = N/M$ denote the sample size on each machine. Let $\{\boldsymbol{X}_i^{(k)}, Y_i^{(k)}\}_{i=1}^n$ be the observations stored on machine $k$, and let $\epsilon_i^{(k)}$ be the corresponding error term, where $1 \leq k \leq M$. On the $k$-th machine, the model is given by

$$Y_i^{(k)} = \boldsymbol{X}_i^{(k)\top} \boldsymbol{\beta}^* + \epsilon_i^{(k)}.$$

The corresponding loss function on the $k$-th machine is

$$\mathcal{L}_k(\boldsymbol{\beta}) = \frac{1}{n(n-1)} \sum_{i \neq j}^n L_h\big((Y_i^{(k)} - Y_j^{(k)}) - (\boldsymbol{X}_i^{(k)} - \boldsymbol{X}_j^{(k)})^\top \boldsymbol{\beta}\big).$$

When the dimension $p$ is much larger than the sample size $n$, we typically consider regularized estimation on each local machine. Specifically, on the $k$-th machine, the regularized estimator is given by

$$\widehat{\boldsymbol{\beta}}_{kh} = \arg\min_{\beta \in \mathbb{R}^p} \Big\{ \mathcal{L}_k(\boldsymbol{\beta}) + \sum_{l=1}^p p_\lambda(\|\beta_l\|) \Big\}. \tag{2.1}$$

Here, $p_\lambda(\cdot)$ is a penalty function and $\lambda$ is the corresponding tuning parameter.

To obtain a final estimator of $\boldsymbol{\beta}^*$, we consider the average of the $\widehat{\boldsymbol{\beta}}_{kh}$'s. However, the regularized estimators $\widehat{\boldsymbol{\beta}}_{kh}$'s are generally biased (Zhang and Huang, 2008; Zhang and Zhang, 2014). Aggregating these estimators does not typically reduce the bias (Zhao et al., 2019). Motivated by the debiased procedures introduced by Zhang and Zhang (2014) and Van de Geer et al. (2014), we also consider the debiased estimator for each machine and conduct statistical inference based on the average of the debiased estimators.

Now we begin to introduce our debiased estimator. Let

$$\boldsymbol{J}_h = \mathbb{E}\big[L_h''(\epsilon_i^{(k)} - \epsilon_j^{(k)})(\boldsymbol{X}_i^{(k)} - \boldsymbol{X}_j^{(k)})(\boldsymbol{X}_i^{(k)} - \boldsymbol{X}_j^{(k)})^\top\big] = 2\mathbb{E}[L_h''(\epsilon_i^{(k)} - \epsilon_j^{(k)})]\boldsymbol{\Sigma}.$$

The sample version of $\boldsymbol{J}_h$ on the $k$-th machine is

$$\widehat{\boldsymbol{J}}_{kh} = \frac{1}{n(n-1)} \sum_{i \neq j}^n L_h''(\widehat{\epsilon}_i^{(k)} - \widehat{\epsilon}_j^{(k)})(\boldsymbol{X}_i^{(k)} - \boldsymbol{X}_j^{(k)})(\boldsymbol{X}_i^{(k)} - \boldsymbol{X}_j^{(k)})^\top.$$

In high dimensional settings, $\widehat{\boldsymbol{J}}_{kh}$ is not invertible. To estimate $\boldsymbol{J}_h^{-1}$, we adopt the method in

Cai et al. (2011) and define $\widehat{\boldsymbol{W}}_{kh}$ as follows:

$$\widehat{\boldsymbol{W}}_{kh} = \arg \min_{\boldsymbol{W} \in \mathbb{R}^{p \times p}} \|\boldsymbol{W}\|_{\infty}$$
$$\text{s.t.} \quad \|\boldsymbol{W}\widehat{\boldsymbol{J}}_{kh} - \boldsymbol{I}\|_{\max} \leq \gamma_n, \tag{2.2}$$

where $\gamma_n$ is a predetermined tuning parameter. From the proof of Cai et al. (2025), it is known that $\|\widehat{\boldsymbol{W}}\widehat{\boldsymbol{J}}_{kh} - \boldsymbol{I}\|_{\max} = O_p(s \log^2(p)/\sqrt{n})$. Thus a suitable choice of $\gamma_n$ is $\gamma_n = O(s \log^2(p)/\sqrt{n})$. Then the debiased CRR on the $k$-th machine is:

$$\widetilde{\boldsymbol{\beta}}_{kh} = \widehat{\boldsymbol{\beta}}_{kh} + \widehat{\boldsymbol{W}}_{kh} \frac{1}{n(n-1)} \sum_{i \neq j}^{n} L_h'(\widehat{\epsilon}_i^{(k)} - \widehat{\epsilon}_j^{(k)})(\boldsymbol{X}_i^{(k)} - \boldsymbol{X}_j^{(k)}), \tag{2.3}$$

where $\widehat{\epsilon}_j^{(k)} = Y_j^{(k)} - (\boldsymbol{X}_j^{(k)})^{\top}\widehat{\boldsymbol{\beta}}_{kh}$. The averaging debiased estimator (ADE) is the average of $\widetilde{\boldsymbol{\beta}}_{kh}$ across all machines:

$$\widetilde{\boldsymbol{\beta}}_{\text{ade}} = \frac{1}{M} \sum_{k=1}^{M} \widetilde{\boldsymbol{\beta}}_{kh}. \tag{2.4}$$

To establish theoretical results for the ADE, some technical assumptions are introduced.

- (A1) Assume that $\widehat{\boldsymbol{\beta}}_{kh}$ satisfies the following conditions for $1 \leq k \leq M$: $\|\widehat{\boldsymbol{\beta}}_{kh} - \boldsymbol{\beta}^*\|_1 = O_p(s\sqrt{\log p/n})$ and $\|\widehat{\boldsymbol{\beta}}_{kh} - \boldsymbol{\beta}^*\|_2 = O_p(\sqrt{s \log p/n})$. Further, assume that the kernel function $K(\cdot)$ is upper bounded, non-negative, symmetric about 0, satisfies $\int_{\mathbb{R}} K(t)\,dt = 1$ and $\int_{-\infty}^{\infty} |t|K(t)\,dt < \infty$. Additionally, assume that $K(\cdot)$ has a bounded derivative.

- (A2) Assume that $\boldsymbol{X}_i$ is a $p$-dimensional sub-Gaussian random vector with $\|\boldsymbol{X}_i\|_{\psi_2} \leq M_0$, and the eigenvalues of the covariance matrix $\boldsymbol{\Sigma}$ satisfy $c_{\boldsymbol{\Sigma}} \leq \lambda_{\min}(\boldsymbol{\Sigma}) \leq \lambda_{\max}(\boldsymbol{\Sigma}) \leq C_{\boldsymbol{\Sigma}}$ for some positive constants $c_{\boldsymbol{\Sigma}}$ and $C_{\boldsymbol{\Sigma}}$.

- (A3) Assume that $\boldsymbol{J}_h^{-1} = (\boldsymbol{\omega}_1, \ldots, \boldsymbol{\omega}_p)^{\top}$ satisfies $\max_{1 \leq i \leq p} \sum_{j=1}^{p} |\omega_{ij}|^q < C_{\omega}$ for some $q \in (0, 1)$ and positive constant $C_{\omega}$.

- (A4) Let $f(\cdot)$ be the probability density function of $\epsilon_i$. Assume that $f(\cdot)$ is continuous and there exists a positive constant $\delta_1$ such that $\inf_{v \in [-\delta_1, \delta_1]} f(v) > 0$. Let $g(\cdot)$ be the probability density function of $\epsilon_i - \epsilon_j$. Assume that $|g(x) - g(y)| \leq L_0 |x - y|, \ \forall x, y \in \mathbb{R}$, for some constant $L_0 > 0$. Further assume that $0 < \underline{g} \leq g(0) \leq \overline{g} < \infty$.

Assumptions (A1)-(A2) are standard conditions on the penalized estimator, kernel function, and covariance matrix, see for instance Zhou et al. (2024). The error bound of $\widehat{\boldsymbol{\beta}}_{kh}$ has been derived by Cai et al. (2025) under some mild assumptions on the density function of $\epsilon_i$, the penalty function $p_{\lambda}(\cdot)$, and the distribution of $\boldsymbol{X}_i$. Assumption (A3) asks $\boldsymbol{J}_h^{-1}$ to be sparse in terms of $\ell_p$ norm of matrix row space. Similar conditions were widely imposed by many authors; see Bickel and Levina (2008), Cai et al. (2011), and Cai et al. (2025). Since the error $\epsilon$ and $\boldsymbol{X}$ are independent, Assumption (A3) can be reduced to sparsity condition on $\boldsymbol{\Sigma}^{-1}$. Assumption (A4) is a mild condition for the density of the error term. Notably no moment condition is required for the error term.

The next two theorems depict the asymptotic behavior of the ADE. We first present the asymptotic representation for the ADE, which shows that $\sqrt{N}(\widetilde{\boldsymbol{\beta}}_{\text{ade}} - \boldsymbol{\beta}^*)$ can be expressed as a high-dimensional U-statistic up to an $o_p(1)$ term.

**Theorem 2.1.** *Let $a = (5 - 4q)/(2 - 2q)$. Under Assumption (A1)-(A4), the ADE satisfies*

$$\|\sqrt{N}(\widetilde{\boldsymbol{\beta}}_{ade} - \boldsymbol{\beta}^*) - \sqrt{N}\boldsymbol{J}_h^{-1}\frac{1}{Mn(n-1)}\sum_{k=1}^{M}\sum_{i \neq j}^{n}L_h'(\epsilon_i^{(k)} - \epsilon_j^{(k)})(\boldsymbol{X}_i^{(k)} - \boldsymbol{X}_j^{(k)})\|_{\infty}$$
$$= O_p(M^{1/2}[s\log^a(p)/\sqrt{n}]^{1-q} + M^{1/2}s^2\log^{5/2}(p)/\sqrt{n}).$$

We can then further obtain the Gaussian approximation result for the ADE presented in the following theorem.

**Theorem 2.2.** *Under Assumption (A1)-(A4), suppose that $M\log^7(pn)n^{r-1} = O(1)$ for some $r \in (0, 1)$, $M^{1/2}[s\log^b(p)/\sqrt{n}]^{1-q} + M^{1/2}s^2\log^3(p)/\sqrt{n} = o(1)$ with $b = (3 - 2q)/(1 - q)$. Then there is a p-dimensional Gaussian random vector $\boldsymbol{Z}_n$ with mean zero and covariance matrix*

$$\boldsymbol{\Sigma}_z = \frac{\mathbb{E}\left[\mathbb{E}^2\left(L_h'(\epsilon_i - \epsilon_j) \mid \epsilon_i\right)\right]}{\left[\mathbb{E}L_h''(\epsilon_i - \epsilon_j)\right]^2}\boldsymbol{\Sigma}^{-1} = \frac{2\mathbb{E}\left[\mathbb{E}^2\left(L_h'(\epsilon_i - \epsilon_j) \mid \epsilon_i\right)\right]}{\mathbb{E}L_h''(\epsilon_i - \epsilon_j)}\boldsymbol{J}_h^{-1},$$

*such that the ADE satisfies*

$$\sup_{t\in\mathbb{R}}\left|\mathbb{P}(\sqrt{N}\|\widetilde{\boldsymbol{\beta}}_{ade} - \boldsymbol{\beta}^*\|_{\infty} \leq t) - \mathbb{P}(\|\boldsymbol{Z}_n\|_{\infty} \leq t)\right| = o(1).$$

Theorem 2.2 establishes the Gaussian approximation for $\sqrt{N}\|\widetilde{\boldsymbol{\beta}}_{\text{ade}} - \boldsymbol{\beta}^*\|_{\infty}$. The theorem implies that $\sqrt{N}(\widetilde{\boldsymbol{\beta}}_{\text{ade}} - \boldsymbol{\beta}^*)$ asymptotically follows a high-dimensional Gaussian distribution, represented by the random vector $\boldsymbol{Z}_n$.

From the above Theorem, we can construct confidence interval for each $\beta_j^*$. Actually on each machine, we can estimate covariance matrix $\boldsymbol{\Sigma}_z$ by

$$\widehat{\boldsymbol{\Sigma}}_z^{(k)} = \frac{2(n-1)\sum_{i=1}^{n}\left\{\sum_{j=1}^{n}L_h'(\widehat{\epsilon}_i^{(k)} - \widehat{\epsilon}_j^{(k)})\right\}^2}{n^2\sum_{i=1}^{n}\sum_{j\neq i}^{n}L_h''(\widehat{\epsilon}_i^{(k)} - \widehat{\epsilon}_j^{(k)})}\widehat{\boldsymbol{W}}_{kh},$$

Let $\boldsymbol{\Sigma}_z = (\sigma_{ij})_{i,j}^p$. We estimate $\sigma_{jj}$ by averaging the results from $M$ machines, denoted as $\widehat{\sigma}_{jj}$. The confidence interval for each $\beta_j^*$ can be constructed as follows:

$$\left\{\widetilde{\boldsymbol{\beta}}_{\text{ade},j} - z_{\alpha/2}\sqrt{\frac{\widehat{\sigma}_{jj}}{N}}, \quad \widetilde{\boldsymbol{\beta}}_{\text{ade},j} + z_{\alpha/2}\sqrt{\frac{\widehat{\sigma}_{jj}}{N}}\right\}. \tag{2.5}$$

Here $z_{\alpha/2}$ is the $\alpha/2$-quantile of the standard normal distribution. We summarize the detailed steps for implementation in Algorithm 1.

**Algorithm 1** Averaging debiased estimator (ADE) for distributed CRR

**Inputs:**

- Distributed data $\{(\boldsymbol{X}_i^{(k)}, Y_i^{(k)})\}_{i=1}^n$ on machines $k = 1, \ldots, M$;
- Bandwidth $h$, penalty parameter $\lambda$, tuning parameter $\gamma_n$, and confidence level $1 - \alpha$;

**For each machine** $k = 1, \ldots, M$**:**

1. Estimate the local penalized CRR estimator

$$\widehat{\boldsymbol{\beta}}_{kh} = \arg\min_{\beta \in \mathbb{R}^p} \left\{ \mathcal{L}_k(\boldsymbol{\beta}) + \sum_{l=1}^p p_\lambda(\|\beta_l\|) \right\}.$$

2. Compute the residuals

$$\widehat{\varepsilon}_i^{(k)} = Y_i^{(k)} - (\boldsymbol{X}_i^{(k)})^\top \widehat{\boldsymbol{\beta}}_{kh}, \qquad i = 1, \ldots, n.$$

3. Compute the local sample version of $\boldsymbol{J}_h$

$$\widehat{\boldsymbol{J}}_{kh} = \frac{1}{n(n-1)} \sum_{i \neq j}^n L_h''(\widehat{\epsilon}_i^{(k)} - \widehat{\epsilon}_j^{(k)}) (\boldsymbol{X}_i^{(k)} - \boldsymbol{X}_j^{(k)}) (\boldsymbol{X}_i^{(k)} - \boldsymbol{X}_j^{(k)})^\top.$$

4. Estimate $\widehat{\boldsymbol{W}}_{kh}$ by

$$\widehat{\boldsymbol{W}}_{kh} = \arg\min_{\boldsymbol{W} \in \mathbb{R}^{p \times p}} \|\boldsymbol{W}\|_\infty \quad \text{s.t.} \quad \|\boldsymbol{W}\widehat{\boldsymbol{J}}_{kh} - \boldsymbol{I}\|_{\max} \leq \gamma_n,$$

5. Compute the local debiased estimator

$$\widetilde{\boldsymbol{\beta}}_{kh} = \widehat{\boldsymbol{\beta}}_{kh} + \widehat{\boldsymbol{W}}_{kh} \frac{1}{n(n-1)} \sum_{i \neq j}^n L_h'(\widehat{\epsilon}_i^{(k)} - \widehat{\epsilon}_j^{(k)}) (\boldsymbol{X}_i^{(k)} - \boldsymbol{X}_j^{(k)}),$$

6. Compute the local variance estimator

$$\widehat{\boldsymbol{\Sigma}}_z^{(k)} = \frac{2(n-1) \sum_{i=1}^n \left\{ \sum_{j=1}^n L_h'(\widehat{\epsilon}_i^{(k)} - \widehat{\epsilon}_j^{(k)}) \right\}^2}{n^2 \sum_{i=1}^n \sum_{j \neq i}^n L_h''(\widehat{\epsilon}_i^{(k)} - \widehat{\epsilon}_j^{(k)})} \widehat{\boldsymbol{W}}_{kh},$$

**Aggregation:**

1. Compute the averaging debiased estimator

$$\widetilde{\boldsymbol{\beta}}_{\mathrm{ade}} = \frac{1}{M} \sum_{k=1}^M \widetilde{\boldsymbol{\beta}}_{kh}.$$

2. For each coordinate $j = 1, \ldots, p$, average the local variance estimates to obtain $\widehat{\sigma}_{jj}$.
3. Construct the $(1 - \alpha)$ confidence interval for $\beta_j^*$:

$$\left\{ \widetilde{\boldsymbol{\beta}}_{\mathrm{ade},j} - z_{\alpha/2}\sqrt{\frac{\widehat{\sigma}_{jj}}{N}}, \quad \widetilde{\boldsymbol{\beta}}_{\mathrm{ade},j} + z_{\alpha/2}\sqrt{\frac{\widehat{\sigma}_{jj}}{N}} \right\}.$$

**Output:** The ADE $\tilde{\boldsymbol{\beta}}_{\mathrm{ade}}$ and the coordinatewise confidence intervals for $\beta_j^*$, $j = 1, \ldots, p$.

Besides the above results, we can also obtain the $\ell_\infty$ error bound of the ADE. To be precise, we have the following theorem.

**Theorem 2.3.** *Under the assumptions in Theorem 2.2, we have that*

$$\|\widetilde{\boldsymbol{\beta}}_{ade} - \boldsymbol{\beta}^*\|_\infty = O_p(\sqrt{\log(p)/N}).$$

Now, we extend ADE to the setting of covariate shift across machines. The theoretical development in Sections 2–3 is presented under the standard assumption that the covariates share the same distribution across machines. In many distributed systems, however, covariate shift is common, i.e., the feature distribution may vary across sites. In this subsection, we show that the averaging debiased estimator (ADE) naturally extends to covariate shift settings with heterogeneous covariance matrices $\{\boldsymbol{\Sigma}_k\}_{k=1}^M$.

Suppose that on machine $k$ ($1 \le k \le M$), we observe i.i.d. samples $\{(\mathbf{X}_i^{(k)}, Y_i^{(k)})\}_{i=1}^n$ from

$$Y_i^{(k)} = (\mathbf{X}_i^{(k)})^\top \boldsymbol{\beta}^* + \epsilon_i^{(k)},$$

where $\boldsymbol{\beta}^* \in \mathbb{R}^p$ is common across machines. We allow $\mathbf{X}_i^{(k)}$ to have site-dependent covariance $\mathrm{Var}(\mathbf{X}_i^{(k)}) = \boldsymbol{\Sigma}_k$, while keeping $\mathbb{E}(\mathbf{X}_i^{(k)}) = \mathbf{0}$. We assume $\mathbf{X}_i^{(k)}$ is independent of $\epsilon_i^{(k)}$ for each $k$, and the error distribution is identical across $k$. Define the population Hessian matrix on machine $k$ as

$$\boldsymbol{J}_{k,h} := \mathbb{E}\Big[ L_h''(\epsilon_i^{(k)} - \epsilon_j^{(k)}) \, (\mathbf{X}_i^{(k)} - \mathbf{X}_j^{(k)})(\mathbf{X}_i^{(k)} - \mathbf{X}_j^{(k)})^\top \Big].$$

Under $\mathbf{X}_i^{(k)} \perp \epsilon_i^{(k)}$ and identical error distributions across sites, we have $\boldsymbol{J}_{k,h} = 2\, \mathbb{E}\big[ L_h''(\epsilon_i - \epsilon_j) \big] \boldsymbol{\Sigma}_k$.

Let $\widehat{\boldsymbol{J}}_{k,h}$, $\widehat{\boldsymbol{W}}_{k,h}$ and $\widetilde{\boldsymbol{\beta}}_{k,h}$ be defined as in (2.2)–(2.3), but with the site-dependent quantities on machine $k$. The *heterogeneity-aware averaging debiased estimator* (HADE) is defined by

$$\widetilde{\boldsymbol{\beta}}_{\text{hade}} := \frac{1}{M} \sum_{k=1}^M \widetilde{\boldsymbol{\beta}}_{k,h}.$$

Note that the construction is identical to ADE in (2.4); the key difference is that $\boldsymbol{J}_{k,h}$ and $\boldsymbol{W}_{k,h}$ are now site-dependent. We keep Assumption (A1) and (A4), and replace (A2)–(A3) by the following heterogeneous versions.

(A2†) For each $k$, $\mathbf{X}_i^{(k)}$ is sub-Gaussian with $\|\mathbf{X}_i^{(k)}\|_{\psi_2} \le M_0$, and $\boldsymbol{\Sigma}_k$ satisfies $c_{\boldsymbol{\Sigma}} \le \lambda_{\min}(\boldsymbol{\Sigma}_k) \le \lambda_{\max}(\boldsymbol{\Sigma}_k) \le C_{\boldsymbol{\Sigma}}$, uniformly over $k$.

(A3†) For each $k$, $\boldsymbol{J}_{k,h}^{-1} = (\boldsymbol{\omega}_1^{(k)}, \dots, \boldsymbol{\omega}_p^{(k)})^\top$ satisfies

$$\max_{1 \le i \le p} \sum_{j=1}^p |\omega_{ij}^{(k)}|^q \le C_\omega,$$

for some $q \in (0,1)$ and constant $C_\omega$ uniformly over $k$.

The following theorem shows that HADE admits the same asymptotic representation rate as ADE, but with a site-dependent leading term.

**Theorem 2.4** (HADE representation under covariate shift)**.** *Let* $a = (5 - 4q)/(2 - 2q)$. *Under*

*Assumptions (A1), (A2†), (A3†) and (A4), HADE satisfies*

$$\left\| \sqrt{N}(\widetilde{\boldsymbol{\beta}}_{\text{hade}} - \boldsymbol{\beta}^*) - \sqrt{N} \frac{1}{Mn(n-1)} \sum_{k=1}^{M} \boldsymbol{J}_{k,h}^{-1} \sum_{i \neq j}^{n} L_h'(\epsilon_i^{(k)} - \epsilon_j^{(k)}) (\mathbf{X}_i^{(k)} - \mathbf{X}_j^{(k)}) \right\|_{\infty}$$
$$= O_p\Big(M^{1/2}[s \log^a(p)/\sqrt{n}]^{1-q} + M^{1/2}s^2 \log^{5/2}(p)/\sqrt{n}\Big).$$

Next we state a high-dimensional Gaussian approximation result. Define

$$\tau_h^2 := \frac{\mathbb{E}\Big[\mathbb{E}^2\big(L_h'(\epsilon_i - \epsilon_j) \mid \epsilon_i\big)\Big]}{\Big(\mathbb{E}L_h''(\epsilon_i - \epsilon_j)\Big)^2}, \qquad \boldsymbol{\Sigma}_{z,\text{hade}} := \tau_h^2 \cdot \frac{1}{M} \sum_{k=1}^{M} \boldsymbol{\Sigma}_k^{-1}.$$

**Theorem 2.5** (HADE Gaussian approximation under covariate shift). *Under Assumptions (A1), (A2†), (A3†) and (A4), suppose that $M \log^7(pn)n^{r-1} = O(1)$ for some $r \in (0,1)$ and*

$$M^{1/2}[s \log^b(p)/\sqrt{n}]^{1-q} + M^{1/2}s^2 \log^3(p)/\sqrt{n} = o(1), \qquad b = (3-2q)/(1-q).$$

*Then there exists a p-dimensional Gaussian random vector $\mathbf{Z}_n$ with mean zero and covariance matrix $\boldsymbol{\Sigma}_{z,\text{hade}}$ such that*

$$\sup_{t \in \mathbb{R}} \Big| \mathbb{P}(\sqrt{N} \|\widetilde{\boldsymbol{\beta}}_{\text{hade}} - \boldsymbol{\beta}^*\|_{\infty} \leq t) - \mathbb{P}(\|\mathbf{Z}_n\|_{\infty} \leq t) \Big| = o(1).$$

Consequently, the confidence interval for $\beta_j^*$ is

$$\Big\{ \widetilde{\beta}_{\text{hade},j} \pm z_{\alpha/2} \sqrt{\widehat{\sigma}_{jj}/N} \Big\},$$

where $\widehat{\sigma}_{jj}$ is obtained by averaging the local variance estimates $\widehat{\sigma}_{jj}^{(k)}$ across machines. To keep the presentation simple, in this paper, we still follow the line of i.i.d machines, assume that data across machines are i.i.d. and share the same distribution.

## 3 Inference based on one-step debiased estimator

To calculate the averaging debiased estimator, we have to calculate each $\widehat{\boldsymbol{W}}_{kh}$ for $1 \leq k \leq M$, which is computationally expensive. To reduce the computational cost, we introduce a one-step debiased estimator (ODE) in this section. The ODE is more computation-efficient, as it only requires the calculation of $\widehat{\boldsymbol{W}}_{1h}$. Specifically, the ODE is defined as follows:

$$\widetilde{\boldsymbol{\beta}}_{\text{ode}} = \widehat{\boldsymbol{\beta}}_{1h} + \widehat{\boldsymbol{W}}_{1h} \frac{1}{Mn(n-1)} \sum_{k=1}^{M} \sum_{i \neq j}^{n} L_h'(\check{\epsilon}_i^{(k)} - \check{\epsilon}_j^{(k)})(\boldsymbol{X}_i^{(k)} - \boldsymbol{X}_j^{(k)}), \qquad (3.1)$$

where $\check{\epsilon}_j^{(k)} = Y_j^{(k)} - (\boldsymbol{X}_j^{(k)})^\top \widehat{\boldsymbol{\beta}}_{1h}$.

We make some discussions about the construction of the ODE. Note that the debiased

estimator on the first machine is given as follows

$$\widetilde{\boldsymbol{\beta}}_1 = \widehat{\boldsymbol{\beta}}_{1h} + \widehat{\boldsymbol{W}}_{1h} \frac{1}{n(n-1)} \sum_{i \neq j}^{n} L_h'(\widehat{\epsilon}_i^{(1)} - \widehat{\epsilon}_j^{(1)})(\boldsymbol{X}_i^{(1)} - \boldsymbol{X}_j^{(1)}),$$

where $\widehat{\epsilon}_j^{(1)} = Y_j^{(1)} - (\boldsymbol{X}_j^{(1)})^\top \widehat{\boldsymbol{\beta}}_{1h}$. However, this debiased estimator cannot achieve root-$N$ convergence rate since there are only $n$ units on the first machine. Instead of only using the gradient information on the first machine, the ODE aggregates all the gradient information from all the machines which makes it achieve the optimal convergence rate. On the other hand, different from the ADE, the ODE requires only one estimator of $\boldsymbol{J}_h^{-1}$, while the ADE requires $M \ \widehat{\boldsymbol{W}}_{kh}$'s. When $M$ is large, the computation can be heavy.

The next two theorems depict the asymptotic behavior for the ODE. Similar to the ADE, we firstly give the asymptotic representation for the ODE, suggesting that $\sqrt{N}(\widetilde{\boldsymbol{\beta}}_{\text{ode}} - \boldsymbol{\beta}^*)$ can be expressed as a high-dimensional U-statistic up to an $o_p(1)$ term.

**Theorem 3.1.** *Let $a = (5 - 4q)/(2 - 2q)$. Under Assumption (A1)-(A3), the ODE satisfies*

$$\|\sqrt{N}(\widetilde{\boldsymbol{\beta}}_{ode} - \boldsymbol{\beta}^*) - \sqrt{N} \boldsymbol{J}_h^{-1} \frac{1}{Mn(n-1)} \sum_{k=1}^{M} \sum_{i \neq j}^{n} L_h'(\epsilon_i^{(k)} - \epsilon_j^{(k)})(\boldsymbol{X}_i^{(k)} - \boldsymbol{X}_j^{(k)})\|_\infty$$

$$= O_p(M^{1/2}[s \log^a(p)/\sqrt{n}]^{1-q} + M^{1/2} s^2 \log^{5/2}(p)/\sqrt{n}).$$

Notice that the asymptotic representation for the ODE is the same as that for the ADE, which means the error bound of the ODE is asymptotically the same as the ADE. But the ODE is more computation-efficient, while the ADE is more communication-efficient. We can further get the Gaussian approximation result for the ODE by the following theorem.

**Theorem 3.2.** *Under Assumption (A1)-(A4), suppose $M \log^7(pn)n^{r-1} = O(1)$ for some $r \in (0,1)$, $M^{1/2}[s \log^b(p)/\sqrt{n}]^{1-q} + M^{1/2} s^2 \log^3(p)/\sqrt{n} = o(1)$ with $b = (3 - 2q)/(1 - q)$. Then there is a $p$-dimensional Gaussian random vector $\boldsymbol{Z}_n$ with mean zero and covariance matrix*

$$\boldsymbol{\Sigma}_z = \frac{\mathbb{E}[\mathbb{E}^2(L_h'(\epsilon_i - \epsilon_j) \mid \epsilon_i)]}{[\mathbb{E}L_h''(\epsilon_i - \epsilon_j)]^2} \boldsymbol{\Sigma}^{-1} = \frac{2\mathbb{E}[\mathbb{E}^2(L_h'(\epsilon_i - \epsilon_j) \mid \epsilon_i)]}{\mathbb{E}L_h''(\epsilon_i - \epsilon_j)} \boldsymbol{J}_h^{-1},$$

*such that the ODE satisfies*

$$\sup_{t \in \mathbb{R}} |\mathbb{P}(\sqrt{N}\|\widetilde{\boldsymbol{\beta}}_{ode} - \boldsymbol{\beta}^*\|_\infty \leq t) - \mathbb{P}(\|\boldsymbol{Z}_n\|_\infty \leq t)| = o(1).$$

Derived from the above theorem, we construct the confidence interval given by the ODE for each $\beta_j^*$ as follows. Now we estimate covariance matrix $\boldsymbol{\Sigma}_z$ only based on the first machine, that is

$$\widehat{\boldsymbol{\Sigma}}_z = \frac{2(n-1)\sum_{i=1}^{n} \left\{ \sum_{j=1}^{n} L_h'(\check{\epsilon}_i^{(1)} - \check{\epsilon}_j^{(1)}) \right\}^2}{n^2 \sum_{i=1}^{n} \sum_{j \neq i}^{n} L_h''(\check{\epsilon}_i^{(1)} - \check{\epsilon}_j^{(1)})} \widehat{\boldsymbol{W}}_{1h},$$

The confidence interval for each $\beta_j^*$ can be constructed as follows:

$$\left\{ \widetilde{\boldsymbol{\beta}}_{\text{ode},j} - z_{\alpha/2} \sqrt{\frac{\widehat{\boldsymbol{\Sigma}}_{z,jj}}{N}}, \quad \widetilde{\boldsymbol{\beta}}_{\text{ode},j} + z_{\alpha/2} \sqrt{\frac{\widehat{\boldsymbol{\Sigma}}_{z,jj}}{N}} \right\}. \tag{3.2}$$

We summarize the detailed steps for implementation in Algorithm 2.

---

**Algorithm 2** One-step debiased estimator (ODE) for distributed CRR

**Inputs:**

- Distributed data $\{(\boldsymbol{X}_i^{(k)}, Y_i^{(k)})\}_{i=1}^n$ on machines $k = 1, \ldots, M$;
- Bandwidth $h$, penalty parameter $\lambda$, and tuning parameter $\gamma_n$, and confidence level $1 - \alpha$;

**Step 1: Initialization on the first machine**

1. Estimate the penalized CRR estimator on the first machine:

$$\widehat{\boldsymbol{\beta}}_{1h} = \arg\min_{\beta \in \mathbb{R}^p} \Big\{ \mathcal{L}_1(\boldsymbol{\beta}) + \sum_{l=1}^p p_\lambda(\|\beta_l\|) \Big\}.$$

2. Compute residuals on the first machine:

$$\widehat{\epsilon}_i^{(1)} = Y_i^{(1)} - \big(\boldsymbol{X}_i^{(1)}\big)^\top \widehat{\boldsymbol{\beta}}_{1h}, \qquad i = 1, \ldots, n.$$

3. Compute

$$\widehat{\boldsymbol{J}}_{1h} = \frac{1}{n(n-1)} \sum_{i \neq j} L_h''\big(\widehat{\varepsilon}_i^{(1)} - \widehat{\varepsilon}_j^{(1)}\big) (\boldsymbol{X}_i^{(1)} - \boldsymbol{X}_j^{(1)})(\boldsymbol{X}_i^{(1)} - \boldsymbol{X}_j^{(1)})^\top.$$

4. Estimate $\widehat{\boldsymbol{W}}_{1h}$ by

$$\widehat{\boldsymbol{W}}_{1h} = \arg\min_{\boldsymbol{W} \in \mathbb{R}^{p \times p}} \|\boldsymbol{W}\|_\infty \quad \text{s.t.} \quad \|\boldsymbol{W}\widehat{\boldsymbol{J}}_{1h} - \boldsymbol{I}\|_{\max} \leq \gamma_n,$$

**Step 2: One-step debiasing**

1. For each machine $k = 1, \ldots, M$, compute residuals using $\widehat{\boldsymbol{\beta}}_{1h}$:

$$\breve{\varepsilon}_i^{(k)} = Y_i^{(k)} - (\boldsymbol{X}_i^{(k)})^\top \widehat{\boldsymbol{\beta}}_{1h}, \qquad i = 1, \ldots, n.$$

2. Compute the ODE

$$\widetilde{\boldsymbol{\beta}}_{\text{ode}} = \widehat{\boldsymbol{\beta}}_{1h} + \widehat{\boldsymbol{W}}_{1h} \frac{1}{Mn(n-1)} \sum_{k=1}^M \sum_{i \neq j} L_h'\big(\breve{\epsilon}_i^{(k)} - \breve{\epsilon}_j^{(k)}\big) \big(\boldsymbol{X}_i^{(k)} - \boldsymbol{X}_j^{(k)}\big),$$

**Step 3: Variance estimation and confidence intervals**

1. Using the first machine, estimate the covariance matrix by

$$\widehat{\boldsymbol{\Sigma}}_z = \frac{2(n-1)\sum_{i=1}^n \big\{ \sum_{j=1}^n L_h'\big(\breve{\epsilon}_i^{(1)} - \breve{\epsilon}_j^{(1)}\big) \big\}^2}{n^2 \sum_{i=1}^n \sum_{j \neq i}^n L_h''\big(\breve{\epsilon}_i^{(1)} - \breve{\epsilon}_j^{(1)}\big)} \widehat{\boldsymbol{W}}_{1h}.$$

2. For each coordinate $j = 1, \ldots, p$, construct the $(1 - \alpha)$ confidence interval

$$\Big\{ \widetilde{\boldsymbol{\beta}}_{\text{ode},j} - z_{\alpha/2} \sqrt{\frac{\widehat{\boldsymbol{\Sigma}}_{z,jj}}{N}}, \quad \widetilde{\boldsymbol{\beta}}_{\text{ode},j} + z_{\alpha/2} \sqrt{\frac{\widehat{\boldsymbol{\Sigma}}_{z,jj}}{N}} \Big\}.$$

**Output:** The ODE $\widetilde{\boldsymbol{\beta}}_{\text{ode}}$ and the coordinatewise confidence intervals for $\beta_j^*$, $j = 1, \ldots, p$.

---

**Remark 1.** *In this paper, following the CRR in Zhou et al. (2024), we treat the bandwidth $h$ as a fixed positive constant. This is reasonable because the convoluted smoothing does not introduce population-level bias, namely $\beta_h^* = \beta^*$ for any $h > 0$. In the distributed setting, the total sample*

*size $N$ is equally split across $M$ local machines, so that the local sample size is $n = N/M$. There-fore, the effect of $M$ enters the theory mainly through the local sample size $n$ and the joint growth conditions involving $M$, $s$, $p$, and $n$. In particular, Theorems 2.1, 2.4, and 3.1 show that the rep-resentation remainders are of order $O_p(M^{1/2}[s\log^a(p)/\sqrt{n}]^{1-q} + M^{1/2}s^2\log^{5/2}(p)/\sqrt{n})$, where $a = (5-4q)/(2-2q)$. Moreover, Theorems 2.2, 2.5, and 3.2 further require $M\log^7(pn)n^{r-1} = O(1)$ for some $r \in (0,1)$, and $M^{1/2}[s\log^b(p)/\sqrt{n}]^{1-q} + M^{1/2}s^2\log^3(p)/\sqrt{n} = o(1)$ with $b = (3-2q)/(1-q)$. Here $h$ is fixed. Therefore, no additional explicit coupling between $h$ and $M$ is imposed in our theory.*

Similarly, the ODE also satisfies the same error bound as the ADE.

**Theorem 3.3.** *Under the assumptions in Theorem 3.2, we have that*

$$\|\widetilde{\boldsymbol{\beta}}_{ode} - \boldsymbol{\beta}^*\|_\infty = O_p\Big(\sqrt{\frac{\log p}{N}}\Big).$$

Finally, note that the debiased estimators (the ADE and the ODE) do not exhibit sparsity. To address this, a hard thresholding approach can be applied to obtain the sparse averaging debiased estimator (SADE) and the sparse one-step debiased estimator (SODE).

Let $c$ be a threshold level. We respectively define SADE and SODE as

$$\widetilde{\beta}^c_{\mathrm{ade},j} = \widetilde{\beta}_{\mathrm{ade},j}\,\mathbb{1}\{|\widetilde{\beta}_{\mathrm{ade},j}| > c\}, \quad \widetilde{\beta}^c_{\mathrm{ode},j} = \widetilde{\beta}_{\mathrm{ode},j}\,\mathbb{1}\{|\widetilde{\beta}_{\mathrm{ode},j}| > c\}.$$

Here $\mathbb{1}\{\cdot\}$ is the indicator function. The advantage of the sparse aggregated estimators is that they reduce both the $\ell_1$ and $\ell_2$ errors of the debiased estimators. We obtain the following result which establishes the estimation error bound of SADE and SODE.

**Theorem 3.4.** *Let $c = c_0\sqrt{\log p/N}$ with a sufficiently large $c_0$, and let $\widetilde{\boldsymbol{\beta}}^c$ be either SADE or SODE. Under the assumptions of Theorem 3.2, we have*

$$\|\widetilde{\boldsymbol{\beta}}^c - \boldsymbol{\beta}^*\|_\infty = O_p\Big(\sqrt{\frac{\log p}{N}}\Big), \quad \|\widetilde{\boldsymbol{\beta}}^c - \boldsymbol{\beta}^*\|_2 = O_p\Big(\sqrt{\frac{s\log p}{N}}\Big), \quad \|\widetilde{\boldsymbol{\beta}}^c - \boldsymbol{\beta}^*\|_1 = O_p\Big(s\sqrt{\frac{\log p}{N}}\Big).$$

When we choose the threshold as $c = c_0\sqrt{\log p/N}$, the sparse debiased estimators achieve the same convergence rate as the global regularized estimator obtained by using all observations in both $\ell_1$ and $\ell_2$ norms.

Next, we present a support recovery result for the sparse debiased estimators.

**Corollary 3.1.** *Let $\mathcal{S} = \{j : \beta_j^* \neq 0\}$. Let $\widetilde{\boldsymbol{\beta}}$ be either $\widetilde{\boldsymbol{\beta}}_{\mathrm{ade}}$ or $\widetilde{\boldsymbol{\beta}}_{\mathrm{ode}}$, and define $(\widetilde{\boldsymbol{\beta}}^c)_j = \widetilde{\beta}_j\mathbb{1}\{|\widetilde{\beta}_j| > c\}$, $j = 1,\ldots,p$. Suppose that the assumptions of Theorem 2.3 hold when $\widetilde{\boldsymbol{\beta}} = \widetilde{\boldsymbol{\beta}}_{\mathrm{ade}}$, and the assumptions of Theorem 3.3 hold when $\widetilde{\boldsymbol{\beta}} = \widetilde{\boldsymbol{\beta}}_{\mathrm{ode}}$. Let $c = c_0\sqrt{\log p/N}$, where $c_0 > 0$ is sufficiently large. Then*

$$\mathbb{P}\{\mathrm{supp}(\widetilde{\boldsymbol{\beta}}^c) \subseteq \mathcal{S}\} \to 1.$$

*If, in addition, $\beta_{\min} := \min_{j\in\mathcal{S}}|\beta_j^*| \geq 2c$, then $\mathbb{P}\{\mathrm{supp}(\widetilde{\boldsymbol{\beta}}^c) = \mathcal{S}\} \to 1$.*

# 4 Numerical simulation

In this section, we conduct numerical simulations to evaluate the finite sample performance of our proposed procedures. We adopt the Epanechnikov kernel as the kernel function, $K(u) = 3(1 - u^2)\mathbb{1}\{-1 \le u \le 1\}/4$, and the loss function is defined by

$$
L_h(u) = \begin{cases} u, & u \ge h, \\ 3u^2h^{-1}/4 - u^4h^{-3}/8 + 3h/8, & -h < u < h, \\ -u, & u \le -h. \end{cases}
$$

Here we fix $h = 1$. We also consider different bandwidths $h = 0.1, 0.5, 1.5$. And the corresponding results are reported in Table S.1 in the Supplementary Material.

We generate data from a linear model as follows:

$$
Y_i = \boldsymbol{X}_i^\top \boldsymbol{\beta}^* + \epsilon_i,
$$

where $\boldsymbol{\beta}^* = (\sqrt{3}, \sqrt{3}, \sqrt{3}, 0, \ldots, 0)^\top$. To evaluate the performance under different true coefficients, we further consider different settings of $\boldsymbol{\beta}$, and the corresponding results are illustrated in Tables S.14-S.17 in the Supplementary Material. The predictor $\boldsymbol{X}_i$ is from the multivariate normal distribution $\mathbf{N}_p(\mathbf{0}_p, \boldsymbol{\Sigma})$. Furthermore, the random error term $\epsilon_i$ is independent of $\boldsymbol{X}_i$. Two scenarios are considered for the distribution of $\epsilon_i$:

- **Scenario 1:** $\epsilon_i$ is simulated from standard normal distribution, i.e. $\epsilon_i \sim N(0, 1)$.

- **Scenario 2:** $\epsilon_i$ follows the mixture gaussian distribution $\tau N(0, 1) + (1 - \tau)N(0, 9)$ with $\tau \sim B(1, 0.95)$.

Additional results for other error distributions, including the t-distribution, the Cauchy distribution, and several asymmetric distributions, are reported in Tables S.6-S.13 in the Supplementary Material. There are two settings for the correlation structure of $\boldsymbol{X}$:

- **Setting 1:** $\boldsymbol{\Sigma}$ is Toeplitz with $\Sigma_{ij} = 0.3^{|i-j|}$.

- **Setting 2:** $\boldsymbol{\Sigma}$ is block diagonal, that is, $\Sigma_{ii} = 1$ for $i = 1, \ldots, p$ and $\Sigma_{ij} = 0.3\mathbb{1}_{\{|i-j|=1\}}$ for $i \neq j$.

Besides, we estimate the unknown parameter $\boldsymbol{\beta}^*$ using the local linear approximation algorithm as described in Algorithm 1 of Zhou et al. (2024).

We compute the centralized debiased estimator (CDE) obtained by a single machine using all observations, ADE, ODE, SADE, and SODE. Our first goal is to assess the accuracy of CDE, ADE, SADE, ODE, and SODE using the $\ell_\infty$ norm and the $\ell_2$ norm under different scenarios and settings. Secondly, we aim to evaluate the confidence intervals based on ADE and ODE under various circumstances. Specifically, we are interested in conducting inference on the coefficients corresponding to certain index set $\mathcal{G}$. Different choices of $\mathcal{G}$ are taken into consideration: $\mathcal{G}_1 = \{1, 2, \ldots, \lfloor p/2 \rfloor\}$, $\mathcal{G}_2 = \{\lfloor p/2 \rfloor + 1, \lfloor p/2 \rfloor + 2, \ldots, p\}$, and $\mathcal{G}_3 = \{1, 2, \ldots, p\}$. We evaluate the average coverage probability (ACP) and the average length (AL) of the 95%

confidence intervals. Specifically, the ACP is defined as $\mathrm{ACP} := \sum_{j \in \mathcal{G}} |\mathcal{G}|^{-1} \mathrm{CP}_j$, where $\mathrm{CP}_j$ denotes the coverage probability of the 95% confidence interval for $\beta_j^*$. Similarly, the AL is given by $\mathrm{AL} := \sum_{j \in \mathcal{G}} |\mathcal{G}|^{-1} \mathrm{AL}_j$, where $\mathrm{AL}_j$ represents the length of the 95% confidence interval for $\beta_j^*$. All results are calculated based on 500 simulation runs.

For the threshold $c$ in sparse aggregated estimators, we consider two approaches. The first approach is to directly set $c = c_0 \sqrt{\log(p)/N}$ with $c_0 = 1.5$. For similar adoption, see also Hou et al. (2023). The other is a rule suggested by Zhao et al. (2019). That is, let $s_{\max} = \max_{1 \le k \le M} \|\widehat{\boldsymbol{\beta}}_{kh}\|_0$, order $|\widetilde{\boldsymbol{\beta}}|_{(1)} \ge \cdots \ge |\widetilde{\boldsymbol{\beta}}|_{(p)}$, and set $c = |\widetilde{\boldsymbol{\beta}}|_{(s_{\max}+1)}$. Here $|\widetilde{\boldsymbol{\beta}}|_{(l)}$ represents the $l$-th largest element of the absolute value version of the ADE or the ODE. The simulation results based on this rule are reported in Figures S.1-S.3 in the Supplementary Material.

First, we consider sample sizes $N =$1000, 2000, 3000, 4000, with $p = 500$ and fix the number of samples in each local machine to be $n =$200. It is seen from Figure 1 that the errors of all the estimators decrease when the total sample size increases. In terms of $\ell_\infty$ error, the differences among all estimators are small. In terms of $\ell_2$ error, SADE and SODE perform better than CDE, ADE, and ODE which are not sparse. In other words, thresholding is effective in reducing the $\ell_2$ error.

In the second set of simulations, we vary $M = 1, 5, 10, 20$ while fixing $N = 2000$ and $p = 500$. Performance generally deteriorates with increasing $M$. Figure 2 shows that, the thresholding is quite effective in reducing the $\ell_2$ error.

In the third set of simulations, we set $N = 2000$, $M = 1, 10$ ($M = 1$ corresponds to the centralized estimator) and $p = 100, 300, 500, 700$. Figure 3 shows the errors of the distributed estimators with $M = 10$. As expected, when the dimension $p$ increases, the errors become larger. Sparse aggregated estimators reduce the $\ell_2$ error, while the $\ell_\infty$ error changes little.

In the final set of simulations, we vary $M = 5, 10, 15, 20$ and $p = 100, 200$ for constructing confidence intervals by fixing the number of samples in each local machine to be $n = 200$. Corresponding numerical results are presented in Tables 1-2. The results in Tables 1-2 provide strong evidence that corroborates our established theoretical results. Actually for all scenarios, settings, and index sets, the coverage probabilities for ADE and ODE are all close to the nominal level 95%. The average lengths of these confidence intervals are also very short.

For comparison, we implement the multiround communication-efficient surrogate smoothed decorrelated score (CSSDS) estimation procedures in Algorithm 1 of Di et al. (2022). Specifically, we set the quantile level to $\tau = 0.5$ and consider $T = 1$ and $T = 2$ in Algorithm 1, and denote the resulting estimators by CSSDS1 and CSSDS2, respectively. The corresponding simulation results are reported in Tables S.2- S.5 in the Supplementary Material.

The simulations are carried out on Windows with R version 4.3.2, using an 8core CPU and 16 GB of memory. For the third set of simulations with $p = 100$, for example, the CDE requires about 31.9 hours to finish all 500 repetitions, while the ADE with $M = 10$ requires about 7.26 hours and the ODE requires about 1.18 hours. This verifies that the ODE is relatively computation-efficient.

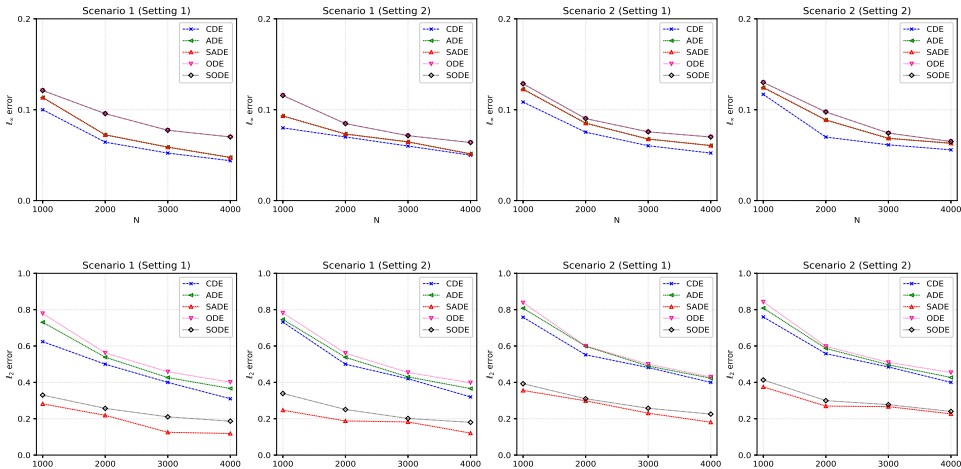

Figure 1: The plot of $\ell_\infty$ and $\ell_2$ errors for $N \in \{1000, 2000, 3000, 4000\}$ under different scenarios and settings. The dashdot line with cross points, dashed line with left triangle points, dashed line with up triangle points, dotted line with down triangle points, and dotted line with diamond points respectively represent the results corresponding to CDE, ADE, SADE, ODE, and SODE.

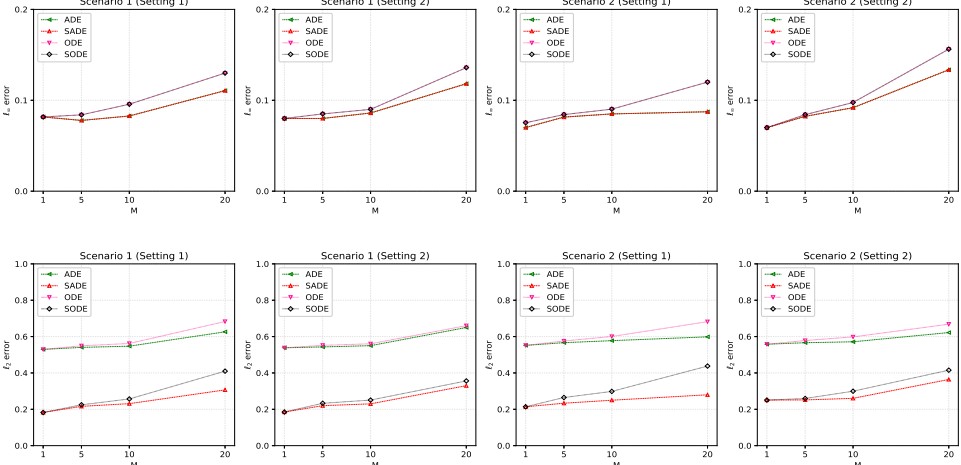

Figure 2: The plot of $\ell_\infty$ and $\ell_2$ errors for $M \in \{1, 5, 10, 20\}$ under different scenarios and settings ($M = 1$ corresponds to the CDE). The dashed line with left triangle points, dashed line with up triangle points, dotted line with down triangle points, and dotted line with diamond points respectively represent the results corresponding to ADE, SADE, ODE, and SODE.

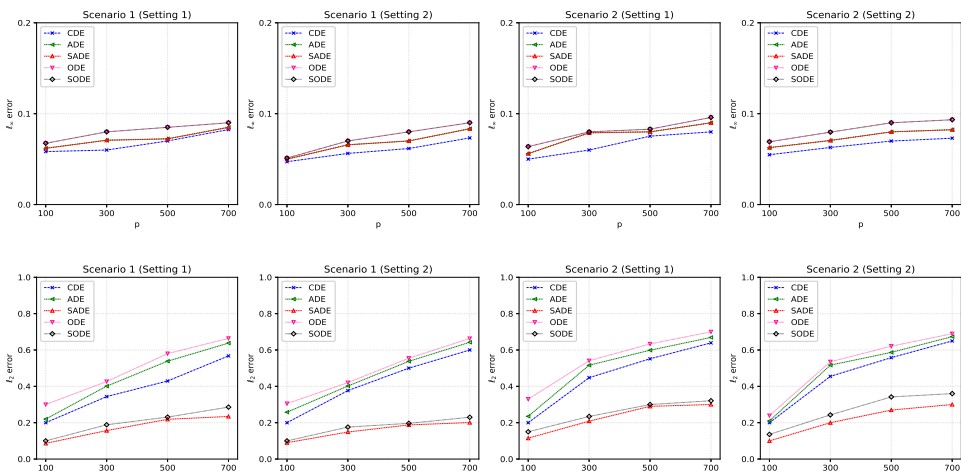

Figure 3: The plot of $\ell_\infty$ and $\ell_2$ errors for $p \in \{100, 300, 500, 700\}$ under different scenarios and settings. The dashdot line with cross points, dashed line with left triangle points, dashed line with up triangle points, dotted line with down triangle points, and dotted line with diamond points respectively represent the results corresponding to CDE, ADE, SADE, ODE, and SODE.

Table 1: Empirical average coverage probability (ACP) and average length (AL) of confidence intervals for ADE and ODE at the 95% level under **Scenario 1** and **Scenario 2** with $p = 100$.

| | | | Scenario 1 | | | | | | Scenario 2 | | | | |
| | | | $\mathcal{G}_1$ | | $\mathcal{G}_2$ | | $\mathcal{G}_3$ | | $\mathcal{G}_1$ | | $\mathcal{G}_2$ | | $\mathcal{G}_3$ | |
| | | $M$ | ACP | AL | ACP | AL | ACP | AL | ACP | AL | ACP | AL | ACP | AL |
|---|---|---|---|---|---|---|---|---|---|---|---|---|---|---|
| **Setting 1** | ADE | 5 | 0.946 | 0.142 | 0.946 | 0.142 | 0.947 | 0.142 | 0.947 | 0.144 | 0.947 | 0.144 | 0.947 | 0.144 |
| | | 10 | 0.944 | 0.101 | 0.944 | 0.101 | 0.945 | 0.101 | 0.949 | 0.102 | 0.949 | 0.102 | 0.947 | 0.102 |
| | | 15 | 0.945 | 0.082 | 0.945 | 0.082 | 0.945 | 0.082 | 0.945 | 0.083 | 0.945 | 0.083 | 0.945 | 0.083 |
| | | 20 | 0.944 | 0.071 | 0.944 | 0.071 | 0.945 | 0.071 | 0.949 | 0.072 | 0.949 | 0.072 | 0.947 | 0.072 |
| | ODE | 5 | 0.944 | 0.133 | 0.944 | 0.133 | 0.944 | 0.133 | 0.940 | 0.133 | 0.948 | 0.133 | 0.944 | 0.133 |
| | | 10 | 0.938 | 0.094 | 0.945 | 0.094 | 0.942 | 0.094 | 0.941 | 0.094 | 0.943 | 0.094 | 0.942 | 0.094 |
| | | 15 | 0.936 | 0.077 | 0.943 | 0.077 | 0.939 | 0.077 | 0.935 | 0.077 | 0.942 | 0.077 | 0.938 | 0.077 |
| | | 20 | 0.930 | 0.066 | 0.943 | 0.066 | 0.937 | 0.066 | 0.933 | 0.067 | 0.944 | 0.067 | 0.939 | 0.067 |
| **Setting 2** | ADE | 5 | 0.943 | 0.151 | 0.943 | 0.151 | 0.942 | 0.151 | 0.941 | 0.152 | 0.941 | 0.152 | 0.942 | 0.152 |
| | | 10 | 0.941 | 0.107 | 0.941 | 0.107 | 0.943 | 0.107 | 0.944 | 0.108 | 0.944 | 0.108 | 0.943 | 0.108 |
| | | 15 | 0.945 | 0.087 | 0.945 | 0.087 | 0.947 | 0.087 | 0.946 | 0.088 | 0.946 | 0.088 | 0.946 | 0.088 |
| | | 20 | 0.944 | 0.075 | 0.944 | 0.075 | 0.944 | 0.075 | 0.943 | 0.076 | 0.943 | 0.076 | 0.943 | 0.076 |
| | ODE | 5 | 0.944 | 0.141 | 0.944 | 0.141 | 0.944 | 0.141 | 0.937 | 0.141 | 0.944 | 0.141 | 0.941 | 0.141 |
| | | 10 | 0.938 | 0.099 | 0.945 | 0.099 | 0.942 | 0.099 | 0.935 | 0.100 | 0.943 | 0.100 | 0.941 | 0.100 |
| | | 15 | 0.935 | 0.081 | 0.947 | 0.081 | 0.941 | 0.081 | 0.934 | 0.081 | 0.944 | 0.081 | 0.941 | 0.081 |
| | | 20 | 0.930 | 0.070 | 0.945 | 0.070 | 0.938 | 0.070 | 0.927 | 0.070 | 0.942 | 0.071 | 0.934 | 0.071 |

Table 2: Empirical average coverage probability (ACP) and average length (AL) of CIs for ADE and ODE at the 95% level under **Scenario 1** and **Scenario 2** with $p = 200$.

| | | $M$ | Scenario 1 $\mathcal{G}_1$ ACP | AL | $\mathcal{G}_2$ ACP | AL | $\mathcal{G}_3$ ACP | AL | Scenario 2 $\mathcal{G}_1$ ACP | AL | $\mathcal{G}_2$ ACP | AL | $\mathcal{G}_3$ ACP | AL |
|---|---|---|---|---|---|---|---|---|---|---|---|---|---|---|
| **Setting 1** | ADE | 5 | 0.944 | 0.144 | 0.944 | 0.144 | 0.947 | 0.142 | 0.944 | 0.146 | 0.944 | 0.146 | 0.943 | 0.146 |
| | | 10 | 0.941 | 0.102 | 0.941 | 0.102 | 0.945 | 0.101 | 0.943 | 0.103 | 0.943 | 0.103 | 0.943 | 0.103 |
| | | 15 | 0.942 | 0.083 | 0.942 | 0.083 | 0.945 | 0.082 | 0.943 | 0.084 | 0.943 | 0.084 | 0.943 | 0.084 |
| | | 20 | 0.941 | 0.072 | 0.941 | 0.072 | 0.945 | 0.071 | 0.942 | 0.073 | 0.942 | 0.073 | 0.942 | 0.073 |
| | ODE | 5 | 0.944 | 0.131 | 0.946 | 0.130 | 0.945 | 0.131 | 0.946 | 0.131 | 0.944 | 0.131 | 0.945 | 0.131 |
| | | 10 | 0.939 | 0.092 | 0.941 | 0.092 | 0.940 | 0.092 | 0.940 | 0.093 | 0.941 | 0.092 | 0.941 | 0.093 |
| | | 15 | 0.939 | 0.075 | 0.942 | 0.075 | 0.939 | 0.075 | 0.940 | 0.076 | 0.941 | 0.076 | 0.941 | 0.076 |
| | | 20 | 0.936 | 0.065 | 0.942 | 0.065 | 0.939 | 0.065 | 0.937 | 0.065 | 0.941 | 0.065 | 0.939 | 0.065 |
| **Setting 2** | ADE | 5 | 0.946 | 0.153 | 0.946 | 0.153 | 0.945 | 0.153 | 0.945 | 0.155 | 0.945 | 0.155 | 0.944 | 0.155 |
| | | 10 | 0.947 | 0.109 | 0.947 | 0.109 | 0.947 | 0.108 | 0.944 | 0.109 | 0.944 | 0.109 | 0.944 | 0.109 |
| | | 15 | 0.946 | 0.089 | 0.946 | 0.089 | 0.944 | 0.089 | 0.944 | 0.089 | 0.944 | 0.089 | 0.944 | 0.089 |
| | | 20 | 0.946 | 0.077 | 0.946 | 0.077 | 0.944 | 0.077 | 0.945 | 0.077 | 0.945 | 0.077 | 0.944 | 0.077 |
| | ODE | 5 | 0.945 | 0.138 | 0.941 | 0.138 | 0.943 | 0.138 | 0.943 | 0.139 | 0.945 | 0.139 | 0.944 | 0.139 |
| | | 10 | 0.943 | 0.098 | 0.945 | 0.098 | 0.944 | 0.098 | 0.941 | 0.098 | 0.945 | 0.098 | 0.944 | 0.098 |
| | | 15 | 0.938 | 0.080 | 0.943 | 0.080 | 0.941 | 0.080 | 0.938 | 0.080 | 0.943 | 0.080 | 0.941 | 0.080 |
| | | 20 | 0.938 | 0.069 | 0.942 | 0.069 | 0.942 | 0.069 | 0.937 | 0.069 | 0.942 | 0.069 | 0.941 | 0.069 |

# 5 Real data analysis

We illustrate our proposed method through studying the Genotype-Tissue Expression(GTEx) data, referred to as GTEx data, which has been analyzed by Li et al. (2022). We focus on the central nervous systems MODULE_137. This dataset is obtained from `https://www.gsea-msigdb.org/gsea/msigdb/cards/MODULE_137.html`. After removing missing values, it includes information on 536 genes of 17,382 observations. The dataset comprises genes measured across 13 brain tissues, such as the Amygdala, Anterior Cingulate Cortex, and Frontal Cortex. Specifically, we are interested in predicting the expression level of gene Dishevelled Segment Polarity Protein 1 (DVL1). DVL1 is a conserved hub of the Wnt pathway. Its activity is regulated by post-translational modifications, including TTLL11-mediated polyglutamylation (Kravec et al., 2024). Perturbation of DVL1 changes $\beta$-catenin signaling and cancer-related traits. Activation of DVL1 promotes stemness and metastasis, whereas knockdown reduces these features, making its expression a clinically relevant and interpretable endpoint (Yin et al., 2021).

We set gene DVL1 as the response variable and the remaining 535 genes as predictors. The large sample size 17,382 makes the computation not sufficient for a central analysis. To this end, the 17,382 observations were randomly divided across $M = 10$ machines. We apply the feature screening method of Li et al. (2012) and retain the top $n/\log(n)$ genes. These selected genes are then used to predict the expression of DVL1.

Our primary objective is to construct confidence intervals for the coefficients of the predictors. Table 3 reports the 95% confidence intervals for the coefficients of three most significant predictors. For ADE, ODE, CSSDS1 and CSSDS2, every confidence interval excludes zero. This suggests a positive association with DVL1 expression. We observe that the gene WDR7 is the most important gene for the expression of DVL1. This gene is also observed in Crummy et al. (2019). Other genes such as MTMR9 and STOML1 have also been highlighted in biomedical research, for example in Qi et al. (2015) and Allen et al. (2020). These consistent observations increase our confidence in the predictors identified by our method and provide useful directions for future study.

Table 3: Confidence intervals at the confidence level 95% in GTEx data.

|        | ADE               | ODE               | CSSDS1          | CSSDS2          |
|--------|-------------------|-------------------|-----------------|-----------------|
| WDR7   | (37.884, 39.268)  | (36.354, 37.691)  | (4.282, 4.700)  | (4.180, 4.605)  |
| MTMR9  | (11.129, 11.772)  | (14.227, 14.899)  | (2.560, 2.801)  | (2.537, 2.782)  |
| STOML1 | (6.322, 6.941)    | (6.196, 6.822)    | (2.310, 2.593)  | (2.266, 2.550)  |

WDR7 denotes WD Repeat Domain 7. MTMR9 denotes Myotubularin Related Protein 9. STOML1 denotes Stomatin Like 1.

# 6 Conclusions and discussions

In this paper, we introduce two debiased estimators for high-dimensional convoluted rank regression in distributed setting. The averaging debiased estimator is a one-shot method and thus

is communication-efficient, while the one-step debiased estimator computes the Hessian matrix only on a single machine and thus is computation-efficient. For these two debiased estimators, we derive their asymptotical representation, Gaussian approximation, and $\ell_\infty$ error bound. Finally, a thresholding approach is involved to construct sparse averaging debiased estimator and sparse one-step debiased estimator, which significantly reduce the $\ell_1$ and $\ell_2$ errors.

There are several possible extensions for future research. Firstly, we may consider investigating distributed inference for high-dimensional semi-parametric convoluted rank regression. Secondly, the development of Byzantine-robust distributed procedure is an important issue. Further in this paper, we assume that there is a central machine which can communicate the estimators or the gradients. However, in practice, decentralized systems are also very common. We will explore these issues in near future.

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
