# Supplementary material for "Distributed inference for high-dimensional convoluted rank regression"

Xinyu Zhang

School of Mathematical Sciences, Capital Normal University, Beijing, 100048, China

Xu Guo

School of Statistics, Beijing Normal University, Beijing, 100875, China

Bingye Yang *

School of Mathematical Sciences, Peking University, Beijing, 100871, China

**Abstract**

This supplementary document includes detailed proofs of the theoretical results and additional simulation studies.

# A    Technical proofs

By a direct calculation, we obtain

$$L_h(u) = u \int_{-u}^{u} K_h(v)\, dv - 2 \int_{-\infty}^{u} v K_h(v)\, dv.$$

Further, we have

$$L_h'(u) = 2 \int_{0}^{u} K_h(v)\, dv, \quad L_h''(u) = 2 K_h(u).$$

The Hessian matrix of $\mathcal{L}_k(\boldsymbol{\beta})$ can be expressed as

$$
\begin{aligned}
&\nabla^2 \mathcal{L}_k(\boldsymbol{\beta}) \\
=&\frac{1}{n(n-1)} \sum_{i \neq j}^{n} L_h''\big(\epsilon_i^{(k)} - \epsilon_j^{(k)} - (\boldsymbol{X}_i^{(k)} - \boldsymbol{X}_j^{(k)})^{\top}(\boldsymbol{\beta} - \boldsymbol{\beta}^*)\big)(\boldsymbol{X}_i^{(k)} - \boldsymbol{X}_j^{(k)})(\boldsymbol{X}_i^{(k)} - \boldsymbol{X}_j^{(k)})^{\top} \\
=&\frac{2}{n(n-1)} \sum_{i \neq j}^{n} K_h\big(\epsilon_i^{(k)} - \epsilon_j^{(k)} - (\boldsymbol{X}_i^{(k)} - \boldsymbol{X}_j^{(k)})^{\top}(\boldsymbol{\beta} - \boldsymbol{\beta}^*)\big)(\boldsymbol{X}_i^{(k)} - \boldsymbol{X}_j^{(k)})(\boldsymbol{X}_i^{(k)} - \boldsymbol{X}_j^{(k)})^{\top}.
\end{aligned}
$$

---

*Corresponding author: byyang25@stu.pku.edu.cn

In this supplementary material, we use $C$, and $C'$ to denote generic constants, whose values can change from line to line.

*Proof of Theorem 2.1.* We can decompose that

$$\sqrt{N}\big(\widetilde{\boldsymbol{\beta}}_{\mathrm{ade}} - \boldsymbol{\beta}^*\big) - \sqrt{N}\boldsymbol{J}_h^{-1}\frac{1}{Mn(n-1)}\sum_{k=1}^M\sum_{i\neq j}^n L_h'\big(\epsilon_i^{(k)} - \epsilon_j^{(k)}\big)\big(\boldsymbol{X}_i^{(k)} - \boldsymbol{X}_j^{(k)}\big)$$

$$=\sqrt{N}\frac{1}{M}\sum_{k=1}^M\big(\widehat{\boldsymbol{\beta}}_{kh} - \boldsymbol{\beta}^*\big) + \sqrt{N}\frac{1}{Mn(n-1)}\sum_{k=1}^M\sum_{i\neq j}^n \widehat{\boldsymbol{W}}_{kh}L_h'\big(\widehat{\epsilon}_i^{(k)} - \widehat{\epsilon}_j^{(k)}\big)\big(\boldsymbol{X}_i^{(k)} - \boldsymbol{X}_j^{(k)}\big)$$

$$-\sqrt{N}\boldsymbol{J}_h^{-1}\frac{1}{Mn(n-1)}\sum_{k=1}^M\sum_{i\neq j}^n L_h'\big(\epsilon_i^{(k)} - \epsilon_j^{(k)}\big)\big(\boldsymbol{X}_i^{(k)} - \boldsymbol{X}_j^{(k)}\big)$$

$$=\sqrt{N}\frac{1}{M}\sum_{k=1}^M\big(\widehat{\boldsymbol{\beta}}_{kh} - \boldsymbol{\beta}^*\big)$$

$$+\sqrt{N}\frac{1}{Mn(n-1)}\sum_{k=1}^M\sum_{i\neq j}^n \big(\widehat{\boldsymbol{W}}_{kh} - \boldsymbol{J}_h^{-1}\big)L_h'\big(\epsilon_i^{(k)} - \epsilon_j^{(k)}\big)\big(\boldsymbol{X}_i^{(k)} - \boldsymbol{X}_j^{(k)}\big)$$

$$+\sqrt{N}\frac{1}{Mn(n-1)}\sum_{k=1}^M\sum_{i\neq j}^n \widehat{\boldsymbol{W}}_{kh}\big[L_h'\big(\widehat{\epsilon}_i^{(k)} - \widehat{\epsilon}_j^{(k)}\big) - L_h'\big(\epsilon_i^{(k)} - \epsilon_j^{(k)}\big)\big]\big(\boldsymbol{X}_i^{(k)} - \boldsymbol{X}_j^{(k)}\big)$$

$$=\sqrt{N}\frac{1}{M}\sum_{k=1}^M\big(\widehat{\boldsymbol{\beta}}_{kh} - \boldsymbol{\beta}^*\big)$$

$$+\sqrt{N}\frac{1}{Mn(n-1)}\sum_{k=1}^M\sum_{i\neq j}^n \big(\widehat{\boldsymbol{W}}_{kh} - \boldsymbol{J}_h^{-1}\big)L_h'\big(\epsilon_i^{(k)} - \epsilon_j^{(k)}\big)\big(\boldsymbol{X}_i^{(k)} - \boldsymbol{X}_j^{(k)}\big)$$

$$-\sqrt{N}\frac{1}{Mn(n-1)}\sum_{k=1}^M\sum_{i\neq j}^n \widehat{\boldsymbol{W}}_{kh}L_h''\big(\widehat{\epsilon}_i^{(k)} - \widehat{\epsilon}_j^{(k)}\big)\big(\boldsymbol{X}_i^{(k)} - \boldsymbol{X}_j^{(k)}\big)\big(\boldsymbol{X}_i^{(k)} - \boldsymbol{X}_j^{(k)}\big)^\top\big(\widehat{\boldsymbol{\beta}}_{kh} - \boldsymbol{\beta}^*\big)$$

$$=\sqrt{N}\frac{1}{M}\sum_{k=1}^M\big(\widehat{\boldsymbol{\beta}}_{kh} - \boldsymbol{\beta}^*\big)$$

$$+\sqrt{N}\frac{1}{Mn(n-1)}\sum_{k=1}^M\sum_{i\neq j}^n \big(\widehat{\boldsymbol{W}}_{kh} - \boldsymbol{J}_h^{-1}\big)L_h'\big(\epsilon_i^{(k)} - \epsilon_j^{(k)}\big)\big(\boldsymbol{X}_i^{(k)} - \boldsymbol{X}_j^{(k)}\big)$$

$$-\sqrt{N}\frac{1}{Mn(n-1)}\sum_{k=1}^M\sum_{i\neq j}^n \widehat{\boldsymbol{W}}_{kh}L_h''\big(\widehat{\epsilon}_i^{(k)} - \widehat{\epsilon}_j^{(k)}\big)\big(\boldsymbol{X}_i^{(k)} - \boldsymbol{X}_j^{(k)}\big)\big(\boldsymbol{X}_i^{(k)} - \boldsymbol{X}_j^{(k)}\big)^\top\big(\widehat{\boldsymbol{\beta}}_{kh} - \boldsymbol{\beta}^*\big)$$

$$-\frac{1}{2}\sqrt{N}\frac{1}{Mn(n-1)}\sum_{k=1}^M\sum_{i\neq j}^n \widehat{\boldsymbol{W}}_{kh}L_h'''\big(\zeta_{ij}\big)\big[\big(\boldsymbol{X}_i^{(k)} - \boldsymbol{X}_j^{(k)}\big)^\top\big(\widehat{\boldsymbol{\beta}}_{kh} - \boldsymbol{\beta}^*\big)\big]^2\big(\boldsymbol{X}_i^{(k)} - \boldsymbol{X}_j^{(k)}\big)$$

$$=\sqrt{N}\frac{1}{M}\sum_{k=1}^M\big(\boldsymbol{I} - \widehat{\boldsymbol{W}}_{kh}\widehat{\boldsymbol{J}}_{kh}\big)\big(\widehat{\boldsymbol{\beta}}_{kh} - \boldsymbol{\beta}^*\big)$$

$$-\frac{1}{2}\sqrt{N}\frac{1}{Mn(n-1)}\sum_{k=1}^M\sum_{i\neq j}^n \widehat{\boldsymbol{W}}_{kh}L_h'''\big(\zeta_{ij}\big)\big[\big(\boldsymbol{X}_i^{(k)} - \boldsymbol{X}_j^{(k)}\big)^\top\big(\widehat{\boldsymbol{\beta}}_{kh} - \boldsymbol{\beta}^*\big)\big]^2\big(\boldsymbol{X}_i^{(k)} - \boldsymbol{X}_j^{(k)}\big)$$

$$+ \sqrt{N} \frac{1}{Mn(n-1)} \sum_{k=1}^{M} \sum_{i \neq j}^{n} (\widehat{\boldsymbol{W}}_{kh} - \boldsymbol{J}_h^{-1}) L_h'\big(\epsilon_i^{(k)} - \epsilon_j^{(k)}\big)\big(\boldsymbol{X}_i^{(k)} - \boldsymbol{X}_j^{(k)}\big)$$

$$:= \mathbf{I} + \mathbf{II} + \mathbf{III}.$$

We next aim to demonstrate that the terms **I**, **II**, and **III** are insignificant under mild conditions. We have that

$$\|\mathbf{I}\|_\infty \leq \frac{\sqrt{N}}{M} \sum_{k=1}^{M} \|(\boldsymbol{I} - \widehat{\boldsymbol{W}}_{kh}\widehat{\boldsymbol{J}}_{kh})(\widehat{\boldsymbol{\beta}}_{kh} - \boldsymbol{\beta}^*)\|_\infty,$$

$$\|\mathbf{II}\|_\infty \leq \frac{\sqrt{N}}{M} \sum_{k=1}^{M} \|\widehat{\boldsymbol{W}}_{kh}\|_\infty \Big\| \frac{1}{2n(n-1)} \sum_{i \neq j}^{n} L_h'''(\zeta_{ij})\big[(\boldsymbol{X}_i^{(k)} - \boldsymbol{X}_j^{(k)})^\top (\widehat{\boldsymbol{\beta}}_{kh} - \boldsymbol{\beta}^*)\big]^2$$
$$\cdot (\boldsymbol{X}_i^{(k)} - \boldsymbol{X}_j^{(k)})\Big\|_\infty,$$

$$\|\mathbf{III}\|_\infty \leq \frac{\sqrt{N}}{M} \sum_{k=1}^{M} \|\widehat{\boldsymbol{W}}_{kh} - \boldsymbol{J}_h^{-1}\|_\infty \Big\| \frac{1}{n(n-1)} \sum_{i \neq j}^{n} L_h'(\epsilon_i^{(k)} - \epsilon_j^{(k)})(\boldsymbol{X}_i^{(k)} - \boldsymbol{X}_j^{(k)})\Big\|_\infty.$$

According to Assumption (A1) and Lemma 4 of Cai et al. (2025) , we have that

$$\|\mathbf{I}\|_\infty \leq \frac{\sqrt{N}}{M} \sum_{k=1}^{M} \|\boldsymbol{I} - \widehat{\boldsymbol{W}}_{kh}\widehat{\boldsymbol{J}}_{kh}\|_{\max} \|\widehat{\boldsymbol{\beta}}_{kh} - \boldsymbol{\beta}^*\|_1$$
$$= O_p\big(M^{1/2} s^2 \log^{5/2}(p)/\sqrt{n}\big).$$

By applying Lemma 4 and the proof of Theorem 2.2 from Cai et al. (2025), we derive that

$$\|\mathbf{II}\|_\infty \lesssim \sqrt{N} \|\widehat{\boldsymbol{W}}_{kh}\|_\infty \Big\| \frac{1}{2Mn(n-1)} \sum_{k=1}^{M} \sum_{i \neq j}^{n} L_h'''(\zeta_{ij})$$
$$\cdot \big[(\boldsymbol{X}_i^{(k)} - \boldsymbol{X}_j^{(k)})^\top (\widehat{\boldsymbol{\beta}}_{kh} - \boldsymbol{\beta}^*)\big]^2 (\boldsymbol{X}_i^{(k)} - \boldsymbol{X}_j^{(k)})\Big\|_\infty$$
$$= O_p\big(M^{1/2} s^2 \log^{5/2}(p)/\sqrt{n}\big).$$

Lemma 3 and Lemma 5 of Cai et al. (2025) guarantee that

$$\|\mathbf{III}\|_\infty \leq \sqrt{N} \|\widehat{\boldsymbol{W}}_{kh} - \boldsymbol{J}_h^{-1}\|_{\max} \Big\| \frac{1}{Mn(n-1)} \sum_{k=1}^{M} \sum_{i \neq j}^{n} L_h'\big(\epsilon_i^{(k)} - \epsilon_j^{(k)}\big)\big(\boldsymbol{X}_i^{(k)} - \boldsymbol{X}_j^{(k)}\big)\Big\|_\infty$$
$$= O_p\Big(M^{1/2}\Big(\frac{s \log^{(5-4q)/(2-2q)}(p)}{\sqrt{n}}\Big)^{1-q}\Big).$$

We summarize

$$\Big\| \sqrt{N}\big(\widetilde{\boldsymbol{\beta}}_{\mathrm{ade}} - \boldsymbol{\beta}^*\big) - \sqrt{N}\,\boldsymbol{J}_h^{-1} \frac{1}{Mn(n-1)} \sum_{k=1}^{M} \sum_{i \neq j}^{n} L_h'\big(\epsilon_i^{(k)} - \epsilon_j^{(k)}\big)\big(\boldsymbol{X}_i^{(k)} - \boldsymbol{X}_j^{(k)}\big)\Big\|_\infty$$
$$= O_p\Big(M^{1/2}\Big\{ \big[s \log^{(5-4q)/(2-2q)}(p)/\sqrt{n}\big]^{1-q} + s^2 \log^{5/2}(p)/\sqrt{n}\Big\}\Big).$$

$\square$

*Proof of Theorem 2.2.* It is easy to see that

$$\sqrt{N}\,\boldsymbol{J}_h^{-1}\frac{1}{Mn(n-1)}\sum_{k=1}^{M}\sum_{i\neq j}^{n}L_h'\big(\epsilon_i^{(k)}-\epsilon_j^{(k)}\big)\big(\boldsymbol{X}_i^{(k)}-\boldsymbol{X}_j^{(k)}\big)$$

$$=\sqrt{n}\,\boldsymbol{J}_h^{-1}\frac{1}{n(n-1)}\sum_{i\neq j}^{n}\Big[\frac{1}{\sqrt{M}}\sum_{k=1}^{M}L_h'\big(\epsilon_i^{(k)}-\epsilon_j^{(k)}\big)\big(\boldsymbol{X}_i^{(k)}-\boldsymbol{X}_j^{(k)}\big)\Big].$$

We next consider the moment and distribution of the above term. Let $\omega_{dl}$ be the $(d,l)$ element of $\boldsymbol{J}_h^{-1}$. To establish asymptotic normality, we construct and calculate as follows. Define

$$\boldsymbol{\xi}_i=\big((\boldsymbol{X}_i^{(1)})^\top,\ \epsilon_i^{(1)},\ (\boldsymbol{X}_i^{(2)})^\top,\ \epsilon_i^{(2)},\ \dots,\ (\boldsymbol{X}_i^{(M)})^\top,\ \epsilon_i^{(M)}\big)^\top,$$

$$f_d(\boldsymbol{\xi}_i,\boldsymbol{\xi}_j)=\frac{1}{\sqrt{M}}\sum_{k=1}^{M}L_h'\big(\epsilon_i^{(k)}-\epsilon_j^{(k)}\big)\sum_{l=1}^{p}\omega_{dl}\big(X_{il}^{(k)}-X_{jl}^{(k)}\big),$$

and $F_d(\boldsymbol{\xi}_i)=\mathbb{E}\big(f_d(\boldsymbol{\xi}_i,\boldsymbol{\xi}_j)\,\big|\,\boldsymbol{\xi}_i\big)$. The random variable $X_{il}^{(k)}$ above denotes the $l$-th element of $\boldsymbol{X}_i^{(k)}$. We calculate the below conditional expectations:

$$F_d(\boldsymbol{\xi}_i)=\frac{1}{\sqrt{M}}\sum_{k=1}^{M}\mathbb{E}\big(L_h'\big(\epsilon_i^{(k)}-\epsilon_j^{(k)}\big)\,\big|\,\epsilon_i^{(k)}\big)\sum_{l=1}^{p}\omega_{dl}X_{il}^{(k)}.$$

Then we have

$$F_d(\boldsymbol{\xi}_i)\,F_t(\boldsymbol{\xi}_i)=\frac{1}{M}\sum_{k_1,k_2,l_1,l_2}\mathbb{E}\big(L_h'\big(\epsilon_i^{(k_1)}-\epsilon_j^{(k_1)}\big)\,\big|\,\epsilon_i^{(k_1)}\big)$$

$$\times\mathbb{E}\big(L_h'\big(\epsilon_i^{(k_2)}-\epsilon_j^{(k_2)}\big)\,\big|\,\epsilon_i^{(k_2)}\big)\,\omega_{dl_1}\omega_{tl_2}X_{il_1}^{(k_1)}X_{il_2}^{(k_2)}$$

$$=\frac{1}{M}\sum_{k_1,k_2=1}^{M}\mathbb{E}\big(L_h'\big(\epsilon_i^{(k_1)}-\epsilon_j^{(k_1)}\big)\,\big|\,\epsilon_i^{(k_1)}\big)$$

$$\times\mathbb{E}\big(L_h'\big(\epsilon_i^{(k_2)}-\epsilon_j^{(k_2)}\big)\,\big|\,\epsilon_i^{(k_2)}\big)\sum_{l_1,l_2=1}^{p}\omega_{dl_1}\omega_{tl_2}X_{il_1}^{(k_1)}X_{il_2}^{(k_2)}.$$

Using the independence of each machine and properties of conditional expectation, we have:

$$\mathbb{E}\big(F_d(\boldsymbol{\xi}_i)F_t(\boldsymbol{\xi}_i)\big)$$

$$=\frac{1}{M}\sum_{k_1,k_2=1}^{M}\mathbb{E}\Big[\mathbb{E}\big(L_h'(\epsilon_i^{(k_1)}-\epsilon_j^{(k_1)})\,\big|\,\epsilon_i^{(k_1)}\big)\mathbb{E}\big(L_h'(\epsilon_i^{(k_2)}-\epsilon_j^{(k_2)})\,\big|\,\epsilon_i^{(k_2)}\big)\Big]$$

$$\cdot\sum_{l_1,l_2=1}^{p}\omega_{dl_1}\mathbb{E}\big[X_{il_1}^{(k_1)}X_{il_2}^{(k_2)}\big]\omega_{tl_2}$$

$$=\frac{1}{M}\sum_{k=1}^{M}\mathbb{E}\Big\{\big[\mathbb{E}\big(L_h'(\epsilon_i^{(k)}-\epsilon_j^{(k)})\,\big|\,\epsilon_i^{(k)}\big)\big]^2\Big\}\sum_{l_1,l_2=1}^{p}\omega_{dl_1}\boldsymbol{\Sigma}\,\omega_{tl_2}$$

$$= \left\{ \mathbb{E} \big[ \mathbb{E} \big( L_h'(\epsilon_i^{(k)} - \epsilon_j^{(k)}) \mid \epsilon_i^{(k)} \big) \big] \right\}^2 \big( \boldsymbol{J}_h^{-1} \boldsymbol{\Sigma} \, \boldsymbol{J}_h^{-1} \big)_{dt}$$

$$= \frac{\left\{ \mathbb{E} \big[ \mathbb{E} \big( L_h'(\epsilon_i^{(k)} - \epsilon_j^{(k)}) \mid \epsilon_i^{(k)} \big) \big] \right\}^2}{4 \big( \mathbb{E} L_h''(\epsilon_i^{(k)} - \epsilon_j^{(k)}) \big)^2} \, \omega_{dt}.$$

Recall that

$$f_d(\boldsymbol{\xi}_i, \boldsymbol{\xi}_j) = \frac{1}{\sqrt{M}} \sum_{k=1}^M L_h'(\epsilon_i^{(k)} - \epsilon_j^{(k)}) \sum_{l=1}^p \omega_{dl} \big( X_{il}^{(k)} - X_{jl}^{(k)} \big).$$

We have

$$\max_{1 \leq d \leq p} \Big\| \frac{1}{\sqrt{M}} \sum_{k=1}^M L_h'(\epsilon_i^{(k)} - \epsilon_j^{(k)}) \sum_{l=1}^p \omega_{dl} \big( X_{il}^{(k)} - X_{jl}^{(k)} \big) \Big\|_{\psi_1}$$

$$\leq \sqrt{M} \max_{1 \leq d \leq p} \Big\| \sum_{l=1}^p \omega_{dl} \big( X_{il}^{(k)} - X_{jl}^{(k)} \big) \Big\|_{\psi_1}$$

$$\leq (\log 2)^{-1/2} \sqrt{M} \max_{1 \leq d \leq p} \Big\| \sum_{l=1}^p \omega_{dl} \big( X_{il}^{(k)} - X_{jl}^{(k)} \big) \Big\|_{\psi_2}$$

$$\leq 2 (\log 2)^{-1/2} \sqrt{M} \, \| X_{il}^{(k)} \|_{\psi_2} \, \| \boldsymbol{J}_h^{-1} \|_\infty = O\big( \sqrt{M} \big).$$

According to the properties of the sub-Gaussian norm, we have

$$\max_{1 \leq d \leq p} \mathbb{E} \big( |f_d(\boldsymbol{\xi}_i, \boldsymbol{\xi}_j)|^m \big) \leq 2 \, \| f_d(\boldsymbol{\xi}_i, \boldsymbol{\xi}_j) \|_{\psi_2}^m \, \Gamma \big( \tfrac{m}{2} + 1 \big).$$

Let $B_n = \sqrt{M} \, \| \boldsymbol{J}_h^{-1} \|_\infty$. According to Corollary 2.2 of Chen (2018), if $M \log^7(pn) \, n^{r-1} = O(1)$ with $r \in (0, 1)$, then there exists a Gaussian random vector $Z_n$ with mean zero and covariance

$$\boldsymbol{\Sigma}_z = \frac{\big\{ \mathbb{E} \big[ \mathbb{E} (L_h'(\epsilon_i^{(k)} - \epsilon_j^{(k)}) \mid \epsilon_i^{(k)}) \big] \big\}^2}{\big[ \mathbb{E} L_h''(\epsilon_i^{(k)} - \epsilon_j^{(k)}) \big]^2} \, \boldsymbol{\Sigma}^{-1},$$

such that

$$\sup_{t \in \mathbb{R}} \Big| \, \mathbb{P} \big( \| \sqrt{N} \, \boldsymbol{J}_h^{-1} \tfrac{1}{2Mn(n-1)} \sum_{k=1}^M \sum_{i \neq j}^n L_h'(\epsilon_i^{(k)} - \epsilon_j^{(k)}) \, ( \boldsymbol{X}_i^{(k)} - \boldsymbol{X}_j^{(k)} ) \|_\infty \leq t \big)$$

$$- \, \mathbb{P} \big( \| \boldsymbol{Z}_n / 2 \|_\infty \leq t \big) \Big| \leq C \, n^{-r/6},$$

where $C > 0$.

Additionally, the assumption $M^{1/2} [s \log^{(3-2q)/(1-q)}(p)/\sqrt{n}]^{1-q} + M^{1/2} s^2 \log^3(p)/\sqrt{n} = o(1)$ yields that the asymptotic representation for the ADE satisfies

$$\Big\| \sqrt{N} \big( \widetilde{\boldsymbol{\beta}}_{\text{ade}} - \boldsymbol{\beta}^* \big) - \sqrt{N} \, \boldsymbol{J}_h^{-1} \tfrac{1}{Mn(n-1)} \sum_{k=1}^M \sum_{i \neq j}^n L_h'(\epsilon_i^{(k)} - \epsilon_j^{(k)}) \big( \boldsymbol{X}_i^{(k)} - \boldsymbol{X}_j^{(k)} \big) \Big\|_\infty$$

$$= o_p \big( \log^{-1/2}(p) \big).$$

Let

$$\boldsymbol{\zeta}_n = \sqrt{N}\, \boldsymbol{J}_h^{-1}\, \frac{1}{Mn(n-1)} \sum_{k=1}^{M} \sum_{i \neq j}^{n} L_h'(\epsilon_i^{(k)} - \epsilon_j^{(k)}) \left( \boldsymbol{X}_i^{(k)} - \boldsymbol{X}_j^{(k)} \right),$$

$$\boldsymbol{\psi}_n = \sqrt{N}\big(\widetilde{\boldsymbol{\beta}}_{\mathrm{ade}} - \boldsymbol{\beta}^*\big) - \sqrt{N}\, \boldsymbol{J}_h^{-1}\, \frac{1}{Mn(n-1)} \sum_{k=1}^{M} \sum_{i \neq j}^{n} L_h'(\epsilon_i^{(k)} - \epsilon_j^{(k)}) \left( \boldsymbol{X}_i^{(k)} - \boldsymbol{X}_j^{(k)} \right).$$

According to Lemma 7 from Cai et al. (2025) and the rate of representation, for any $\delta > 0$,

$$\sup_{t \in \mathbb{R}} \big| \mathbb{P}\big(\sqrt{N}\, \|\widetilde{\boldsymbol{\beta}}_{\mathrm{ade}} - \boldsymbol{\beta}^*\|_\infty \geq t\big) - \mathbb{P}\big(\|\boldsymbol{Z}_n\|_\infty \geq t\big) \big|$$

$$= \sup_{t \in \mathbb{R}} \big| \mathbb{P}\big(\|\boldsymbol{\zeta}_n + \boldsymbol{\psi}_n\|_\infty \geq t\big) - \mathbb{P}\big(\|\boldsymbol{Z}_n\|_\infty \geq t\big) \big|$$

$$\leq \mathbb{P}\big(\|\boldsymbol{\psi}_n\|_\infty > \delta \log^{-1/2}(p)\big) + \sup_{t \in \mathbb{R}} \mathbb{P}\big(|\|\boldsymbol{Z}_n\|_\infty - t| < \delta \log^{-1/2}(p)\big) + C\, n^{-r/6}$$

$$\leq \mathbb{P}\big(\|\boldsymbol{\psi}_n\|_\infty > \delta \log^{-1/2}(p)\big) + C'\, \delta + C\, n^{-r/6},$$

where $C' > 0$ and $C > 0$.

By taking a small $\delta$, we have

$$\sup_{t \in \mathbb{R}} \big| \mathbb{P}\big(\sqrt{N}\, \|\widetilde{\boldsymbol{\beta}}_{\mathrm{ade}} - \boldsymbol{\beta}^*\|_\infty \geq t\big) - \mathbb{P}\big(\|\boldsymbol{Z}_n\|_\infty \geq t\big) \big| = o(1).$$

$\square$

*Proof of Theorem 2.3.* Recall that $\boldsymbol{\Sigma}_z = (\sigma_{ij})_{i,j=1}^{p}$. By assumption (A2), for some $0 < b_1 < b_2$, we have $b_1 < \sigma_{jj} < b_2$ for all $1 \leq j \leq p$. Denote $\boldsymbol{Z}_n = (Z_1, \ldots, Z_p)^\top$. Notice that

$$\mathbb{P}\big(\|\boldsymbol{Z}_n\|_\infty \geq t\big) = \mathbb{P}\big(\max_{1 \leq i \leq p} |Z_i| \geq t\big) = \mathbb{P}\Big(\bigcup_{i=1}^{p} \{|Z_i| \geq t\}\Big) \leq \sum_{i=1}^{p} \mathbb{P}\big(|Z_i| \geq t\big).$$

Then, by Chernoff bound, we have

$$\mathbb{P}\big(\|\boldsymbol{Z}_n\|_\infty \geq t\big) \leq \sum_{i=1}^{p} 2 \exp\big(-t^2/2\sigma_{jj}\big) \leq 2p \exp\big(-t^2/2b_1\big) = \exp\Big(\frac{-t^2 + 2b_1 \log(2p)}{2b_1}\Big).$$

Let $t = C\sqrt{\log p}$, where $C > 0$, we have

$$\mathbb{P}\big(\|\boldsymbol{Z}_n\|_\infty \leq C\sqrt{\log p}\big) \geq 1 - 1/p.$$

Then Theorem 2.2 ensures that

$$\mathbb{P}\big(\sqrt{N}\, \|\widetilde{\boldsymbol{\beta}}_{\mathrm{ade}} - \boldsymbol{\beta}^*\|_\infty \leq C\sqrt{\log p}\big) \geq 1 - 1/p.$$

Thus, we have

$$\|\widetilde{\boldsymbol{\beta}}_{\mathrm{ade}} - \boldsymbol{\beta}^*\|_\infty = O_p\Big(\sqrt{\frac{\log p}{N}}\Big).$$

$\square$

*Proof of Theorem 2.4.* The proof follows the same decomposition as Theorem 2.1, with the key difference that the population Hessian is site-dependent. Write

$$\widetilde{\boldsymbol{\beta}}_{\text{hade}} = \frac{1}{M} \sum_{k=1}^{M} \widetilde{\boldsymbol{\beta}}_{k,h}, \qquad \widetilde{\boldsymbol{\beta}}_{k,h} = \widehat{\boldsymbol{\beta}}_{k,h} + \widehat{\boldsymbol{W}}_{k,h} \frac{1}{n(n-1)} \sum_{i \neq j}^{n} L_h'\big(\widehat{\epsilon}_i^{(k)} - \widehat{\epsilon}_j^{(k)}\big) (\mathbf{X}_i^{(k)} - \mathbf{X}_j^{(k)}).$$

A second-order Taylor expansion yields

$$L_h'\big(\widehat{\epsilon}_i^{(k)} - \widehat{\epsilon}_j^{(k)}\big) = L_h'\big(\epsilon_i^{(k)} - \epsilon_j^{(k)}\big) - L_h''(\widehat{\epsilon}_i^{(k)} - \widehat{\epsilon}_j^{(k)})(\mathbf{X}_i^{(k)} - \mathbf{X}_j^{(k)})^\top(\widehat{\boldsymbol{\beta}}_{k,h} - \boldsymbol{\beta}^*) - \frac{1}{2}L_h'''(\zeta_{ij}^{(k)})\Delta_{ij}^{(k)2},$$

where $\Delta_{ij}^{(k)} = (\mathbf{X}_i^{(k)} - \mathbf{X}_j^{(k)})^\top(\widehat{\boldsymbol{\beta}}_{k,h} - \boldsymbol{\beta}^*)$ and $\zeta_{ij}^{(k)}$ lies between $\epsilon_i^{(k)} - \epsilon_j^{(k)}$ and $\widehat{\epsilon}_i^{(k)} - \widehat{\epsilon}_j^{(k)}$. Plugging the expansion into $\widetilde{\boldsymbol{\beta}}_{k,h}$ and rearranging terms gives

$$\widetilde{\boldsymbol{\beta}}_{k,h} - \boldsymbol{\beta}^* = (\mathbf{I} - \widehat{\boldsymbol{W}}_{k,h}\widehat{\boldsymbol{J}}_{k,h})(\widehat{\boldsymbol{\beta}}_{k,h} - \boldsymbol{\beta}^*) + \widehat{\boldsymbol{W}}_{k,h}\frac{1}{n(n-1)} \sum_{i \neq j}^{n} L_h'\big(\epsilon_i^{(k)} - \epsilon_j^{(k)}\big)(\mathbf{X}_i^{(k)} - \mathbf{X}_j^{(k)}) + \boldsymbol{R}_{k,h},$$

where $\|\boldsymbol{R}_{k,h}\|_\infty$ is controlled by the third-derivative remainder and satisfies $\|\boldsymbol{R}_{k,h}\|_\infty = O_p(s^2 \log^{5/2}(p)/n)$ under Assumption (A1) and the same argument used in Cai et al. (2025). Therefore,

$$\sqrt{N}\big(\widetilde{\boldsymbol{\beta}}_{\text{hade}} - \boldsymbol{\beta}^*\big) - \sqrt{N}\,\frac{1}{Mn(n-1)} \sum_{k=1}^{M} \boldsymbol{J}_{k,h}^{-1} \sum_{i \neq j}^{n} L_h'\big(\epsilon_i^{(k)} - \epsilon_j^{(k)}\big)(\mathbf{X}_i^{(k)} - \mathbf{X}_j^{(k)})$$

$$= \frac{\sqrt{N}}{M} \sum_{k=1}^{M} (\mathbf{I} - \widehat{\boldsymbol{W}}_{k,h}\widehat{\boldsymbol{J}}_{k,h})(\widehat{\boldsymbol{\beta}}_{k,h} - \boldsymbol{\beta}^*) + \frac{\sqrt{N}}{M} \sum_{k=1}^{M} \boldsymbol{R}_{k,h}$$

$$+ \sqrt{N}\,\frac{1}{Mn(n-1)} \sum_{k=1}^{M} (\widehat{\boldsymbol{W}}_{k,h} - \boldsymbol{J}_{k,h}^{-1}) \sum_{i \neq j}^{n} L_h'\big(\epsilon_i^{(k)} - \epsilon_j^{(k)}\big)(\mathbf{X}_i^{(k)} - \mathbf{X}_j^{(k)}).$$

The first two terms are bounded by

$$O_p\Big(\sqrt{N}\,\gamma_n\,s\sqrt{\log p/n}\Big) = O_p\big(M^{1/2}s^2 \log^{5/2}(p)/\sqrt{n}\big),$$

using $\|\mathbf{I} - \widehat{\boldsymbol{W}}_{k,h}\widehat{\boldsymbol{J}}_{k,h}\|_{\max} \leq \gamma_n$ and Assumption (A1). The last term is bounded by

$$O_p\Big(\sqrt{N}\,\|\widehat{\boldsymbol{W}}_{k,h} - \boldsymbol{J}_{k,h}^{-1}\|_\infty\Big) = O_p\Big(M^{1/2}\big[s \log^a(p)/\sqrt{n}\big]^{1-q}\Big),$$

using the same CLIME-type bound as in Cai et al. (2025), which applies to each machine under (A2$^\dagger$)–(A3$^\dagger$). Combining the bounds completes the proof. $\square$

*Proof of Theorem 2.5.* By Theorem 2.4, it suffices to establish Gaussian approximation for the leading term

$$\boldsymbol{T}_n := \sqrt{N}\,\frac{1}{Mn(n-1)} \sum_{k=1}^{M} \boldsymbol{J}_{k,h}^{-1} \sum_{i \neq j}^{n} L_h'\big(\epsilon_i^{(k)} - \epsilon_j^{(k)}\big)(\mathbf{X}_i^{(k)} - \mathbf{X}_j^{(k)}).$$

Rewrite $\boldsymbol{T}_n$ as a $U$-statistic in i.i.d. "stacked" observations. Define

$$\boldsymbol{\xi}_i = \left( (\mathbf{X}_i^{(1)})^\top, \epsilon_i^{(1)}, \ldots, (\mathbf{X}_i^{(M)})^\top, \epsilon_i^{(M)} \right)^\top, \qquad i = 1, \ldots, n.$$

Since each site has i.i.d. samples and sites are independent, $\{\boldsymbol{\xi}_i\}_{i=1}^n$ are i.i.d. across $i$ even if the marginal distributions differ across machines. For each coordinate $d \in \{1, \ldots, p\}$, define the symmetric kernel

$$f_d(\boldsymbol{\xi}_i, \boldsymbol{\xi}_j) = \frac{1}{\sqrt{M}} \sum_{k=1}^M L_h'\left(\epsilon_i^{(k)} - \epsilon_j^{(k)}\right) \sum_{l=1}^p \omega_{dl}^{(k)}\left(X_{il}^{(k)} - X_{jl}^{(k)}\right),$$

where $\omega_{dl}^{(k)}$ is the $(d, l)$ element of $\boldsymbol{J}_{k,h}^{-1}$. Then $\boldsymbol{T}_n$ can be written as

$$\boldsymbol{T}_n = \frac{\sqrt{n}}{n(n-1)} \sum_{i \neq j}^n \left( f_1(\boldsymbol{\xi}_i, \boldsymbol{\xi}_j), \ldots, f_p(\boldsymbol{\xi}_i, \boldsymbol{\xi}_j) \right)^\top.$$

Let $F_d(\boldsymbol{\xi}_i) = \mathbb{E}(f_d(\boldsymbol{\xi}_i, \boldsymbol{\xi}_j) \mid \boldsymbol{\xi}_i)$. Using $\mathbb{E}(\mathbf{X}_i^{(k)}) = \mathbf{0}$ and independence across sites, cross-machine terms vanish and we obtain

$$\mathbb{E}\left[ F_d(\boldsymbol{\xi}_i) F_t(\boldsymbol{\xi}_i) \right] = \frac{1}{M} \sum_{k=1}^M \mathbb{E}\left[ \mathbb{E}^2\left( L_h'(\epsilon_i - \epsilon_j) \mid \epsilon_i \right) \right] \boldsymbol{\omega}_d^{(k)\top} \boldsymbol{\Sigma}_k \boldsymbol{\omega}_t^{(k)},$$

where $\boldsymbol{\omega}_d^{(k)}$ denotes the $d$-th row of $\boldsymbol{J}_{k,h}^{-1}$. Since $\boldsymbol{J}_{k,h} = 2\,\mathbb{E}[L_h''(\epsilon_i - \epsilon_j)]\,\boldsymbol{\Sigma}_k$, we have $\boldsymbol{\omega}_d^{(k)\top} \boldsymbol{\Sigma}_k \boldsymbol{\omega}_t^{(k)} = (4\,\mathbb{E}^2 L_h'')^{-1}(\boldsymbol{\Sigma}_k^{-1})_{dt}$, and therefore the covariance matrix of the Hájek projection equals

$$4\mathbb{E}\left[ F(\boldsymbol{\xi}_i) F(\boldsymbol{\xi}_i)^\top \right] = \tau_h^2 \cdot \frac{1}{M} \sum_{k=1}^M \boldsymbol{\Sigma}_k^{-1} = \boldsymbol{\Sigma}_{z,\text{hade}}.$$

Finally, under Assumptions (A2$^\dagger$)–(A3$^\dagger$), each coordinate of $f_d(\boldsymbol{\xi}_i, \boldsymbol{\xi}_j)$ has uniformly bounded sub-exponential norm (up to $\sqrt{M}$), so Corollary 2.2 of Chen (2018) yields the desired high-dimensional Gaussian approximation for $\|\boldsymbol{T}_n\|_\infty$. Combining this with the representation error control in Theorem 2.4 completes the proof. □

*Proof of Theorem 3.1.* By a direct calculation, we have

$$\widetilde{\boldsymbol{\beta}}_{\text{ode}} - \boldsymbol{\beta}^*$$

$$= \widehat{\boldsymbol{\beta}}_{1h} - \boldsymbol{\beta}^* + \widehat{\boldsymbol{W}}_{1h} \frac{1}{Mn(n-1)} \sum_{k=1}^M \sum_{i \neq j}^n L_h'\left(\breve{\epsilon}_i^{(k)} - \breve{\epsilon}_j^{(k)}\right)\left(\boldsymbol{X}_i^{(k)} - \boldsymbol{X}_j^{(k)}\right).$$

We can decompose that

$$\sqrt{N}\left(\widetilde{\boldsymbol{\beta}}_{\text{ode}} - \boldsymbol{\beta}^*\right) - \sqrt{N}\boldsymbol{J}_h^{-1} \frac{1}{Mn(n-1)} \sum_{k=1}^M \sum_{i \neq j}^n L_h'\left(\epsilon_i^{(k)} - \epsilon_j^{(k)}\right)\left(\boldsymbol{X}_i^{(k)} - \boldsymbol{X}_j^{(k)}\right)$$

$$= \sqrt{N}\big(\widehat{\boldsymbol{\beta}}_{1h} - \boldsymbol{\beta}^*\big) + \sqrt{N}\widehat{\boldsymbol{W}}_{1h}\frac{1}{Mn(n-1)}\sum_{k=1}^{M}\sum_{i\neq j}^{n} L_h'\big(\breve{\epsilon}_i^{(k)} - \breve{\epsilon}_j^{(k)}\big)\big(\boldsymbol{X}_i^{(k)} - \boldsymbol{X}_j^{(k)}\big)$$

$$- \sqrt{N}\boldsymbol{J}_h^{-1}\frac{1}{Mn(n-1)}\sum_{k=1}^{M}\sum_{i\neq j}^{n} L_h'\big(\epsilon_i^{(k)} - \epsilon_j^{(k)}\big)\big(\boldsymbol{X}_i^{(k)} - \boldsymbol{X}_j^{(k)}\big)$$

$$= \sqrt{N}\big(\widehat{\boldsymbol{\beta}}_{1h} - \boldsymbol{\beta}^*\big) + \sqrt{N}\big(\widehat{\boldsymbol{W}}_{1h} - \boldsymbol{J}_h^{-1}\big)\frac{1}{Mn(n-1)}\sum_{k=1}^{M}\sum_{i\neq j}^{n} L_h'\big(\epsilon_i^{(k)} - \epsilon_j^{(k)}\big)\big(\boldsymbol{X}_i^{(k)} - \boldsymbol{X}_j^{(k)}\big)$$

$$+ \sqrt{N}\widehat{\boldsymbol{W}}_{1h}\frac{1}{Mn(n-1)}\sum_{k=1}^{M}\sum_{i\neq j}^{n}\Big[L_h'\big(\breve{\epsilon}_i^{(k)} - \breve{\epsilon}_j^{(k)}\big) - L_h'\big(\epsilon_i^{(k)} - \epsilon_j^{(k)}\big)\Big]\big(\boldsymbol{X}_i^{(k)} - \boldsymbol{X}_j^{(k)}\big)$$

$$= \sqrt{N}\big(\widehat{\boldsymbol{\beta}}_{1h} - \boldsymbol{\beta}^*\big) + \sqrt{N}\big(\widehat{\boldsymbol{W}}_{1h} - \boldsymbol{J}_h^{-1}\big)\frac{1}{Mn(n-1)}\sum_{k=1}^{M}\sum_{i\neq j}^{n} L_h'\big(\epsilon_i^{(k)} - \epsilon_j^{(k)}\big)\big(\boldsymbol{X}_i^{(k)} - \boldsymbol{X}_j^{(k)}\big)$$

$$- \sqrt{N}\widehat{\boldsymbol{W}}_{1h}\frac{1}{Mn(n-1)}\sum_{k=1}^{M}\sum_{i\neq j}^{n} L_h''\big(\breve{\epsilon}_i^{(k)} - \breve{\epsilon}_j^{(k)}\big)\big(\boldsymbol{X}_i^{(k)} - \boldsymbol{X}_j^{(k)}\big)\big(\boldsymbol{X}_i^{(k)} - \boldsymbol{X}_j^{(k)}\big)^\top\big(\widehat{\boldsymbol{\beta}}_{1h} - \boldsymbol{\beta}^*\big)$$

$$- \frac{1}{2}\sqrt{N}\widehat{\boldsymbol{W}}_{1h}\frac{1}{Mn(n-1)}\sum_{k=1}^{M}\sum_{i\neq j}^{n} L_h'''(\zeta_{ij})\Big[\big(\boldsymbol{X}_i^{(k)} - \boldsymbol{X}_j^{(k)}\big)^\top\big(\widehat{\boldsymbol{\beta}}_{1h} - \boldsymbol{\beta}^*\big)\Big]^2\big(\boldsymbol{X}_i^{(k)} - \boldsymbol{X}_j^{(k)}\big)$$

$$= \sqrt{N}\Big(\boldsymbol{I} - \widehat{\boldsymbol{W}}_{1h}\frac{1}{M}\sum_{k=1}^{M}\widehat{\boldsymbol{J}}_{kh}\Big)\big(\widehat{\boldsymbol{\beta}}_{1h} - \boldsymbol{\beta}^*\big)$$

$$- \frac{1}{2}\sqrt{N}\widehat{\boldsymbol{W}}_{1h}\frac{1}{Mn(n-1)}\sum_{k=1}^{M}\sum_{i\neq j}^{n} L_h'''(\zeta_{ij})\Big[\big(\boldsymbol{X}_i^{(k)} - \boldsymbol{X}_j^{(k)}\big)^\top\big(\widehat{\boldsymbol{\beta}}_{1h} - \boldsymbol{\beta}^*\big)\Big]^2\big(\boldsymbol{X}_i^{(k)} - \boldsymbol{X}_j^{(k)}\big)$$

$$+ \sqrt{N}\big(\widehat{\boldsymbol{W}}_{1h} - \boldsymbol{J}_h^{-1}\big)\frac{1}{Mn(n-1)}\sum_{k=1}^{M}\sum_{i\neq j}^{n} L_h'\big(\epsilon_i^{(k)} - \epsilon_j^{(k)}\big)\big(\boldsymbol{X}_i^{(k)} - \boldsymbol{X}_j^{(k)}\big)$$

$$:= \mathbf{IV} + \mathbf{V} + \mathbf{VI}.$$

Next, we aim to show that the terms $\mathbf{IV}$, $\mathbf{V}$, and $\mathbf{VI}$ are negligible under mild conditions. Notice that

$$\boldsymbol{I} - \widehat{\boldsymbol{W}}_{1h}\frac{1}{M}\sum_{k=1}^{M}\widehat{\boldsymbol{J}}_{kh}$$

$$= \frac{1}{M}\sum_{k=1}^{M}\big(\boldsymbol{I} - \widehat{\boldsymbol{W}}_{1h}\widehat{\boldsymbol{J}}_{kh}\big)$$

$$= \frac{1}{M}\sum_{k=1}^{M}\Big[\big(\boldsymbol{I} - \widehat{\boldsymbol{W}}_{1h}\boldsymbol{J}_h\big) - \widehat{\boldsymbol{W}}_{1h}\big(\widehat{\boldsymbol{J}}_{kh} - \boldsymbol{J}_h\big)\Big].$$

By Lemma 4 of Cai et al. (2025), we obtain that

$$\|\boldsymbol{I} - \widehat{\boldsymbol{W}}_{1h}\frac{1}{M}\sum_{k=1}^{M}\widehat{\boldsymbol{J}}_{kh}\|_{\max}$$

$$\leq \frac{1}{M} \sum_{k=1}^{M} \Big[ \|\boldsymbol{I} - \widehat{\boldsymbol{W}}_{1h} \boldsymbol{J}_h\|_{\max} + \|\widehat{\boldsymbol{W}}_{1h}\|_1 \|\widehat{\boldsymbol{J}}_{kh} - \boldsymbol{J}_h\|_{\max} \Big]$$

$$= O\big(s \log^2 p / \sqrt{n}\big).$$

Then, for term **IV**, we have

$$\|\mathbf{IV}\|_\infty \leq \sqrt{N} \, \|\boldsymbol{I} - \widehat{\boldsymbol{W}}_{1h} \frac{1}{M} \sum_{k=1}^{M} \widehat{\boldsymbol{J}}_{kh}\|_{\max} \|\widehat{\boldsymbol{\beta}}_{1h} - \boldsymbol{\beta}^*\|_1$$

$$= O_p\big(M^{1/2} s^2 \log^{5/2} p / \sqrt{n}\big).$$

By taking Lemma 4 and the proof of Theorem 2.2 from Cai et al. (2025), we have that

$$\|\mathbf{V}\|_\infty \leq \sqrt{N} \, \|\widehat{\boldsymbol{W}}_{1h}\|_\infty \Big\| \frac{1}{2Mn(n-1)} \sum_{k=1}^{M} \sum_{i \neq j}^{n} L_h'''(\zeta_{ij}) \Big[ \big(\boldsymbol{X}_i^{(k)} - \boldsymbol{X}_j^{(k)}\big)^\top \big(\widehat{\boldsymbol{\beta}}_{1h} - \boldsymbol{\beta}^*\big) \Big]^2$$

$$\cdot \big(\boldsymbol{X}_i^{(k)} - \boldsymbol{X}_j^{(k)}\big) \Big\|_\infty$$

$$= O_p\big(M^{1/2} s^2 \log^{5/2} p / \sqrt{n}\big).$$

Lemma 3 and Lemma 5 of Cai et al. (2025) ensure that

$$\|\mathbf{VI}\|_\infty \leq \sqrt{N} \, \|\widehat{\boldsymbol{W}}_{1h} - \boldsymbol{J}_h^{-1}\|_\infty \Big\| \frac{1}{Mn(n-1)} \sum_{k=1}^{M} \sum_{i \neq j}^{n} L_h'\big(\epsilon_i^{(k)} - \epsilon_j^{(k)}\big) \big(\boldsymbol{X}_i^{(k)} - \boldsymbol{X}_j^{(k)}\big) \Big\|_\infty$$

$$= O_p\big(M^{1/2} \big[s \log^{(5-4q)/(2-2q)} p / \sqrt{n}\big]^{1-q}\big).$$

In conclusion, we have

$$\Big\| \sqrt{N} \big(\widetilde{\boldsymbol{\beta}}_{\mathrm{ode}} - \boldsymbol{\beta}^*\big) - \sqrt{N} \, \boldsymbol{J}_h^{-1} \frac{1}{Mn(n-1)} \sum_{k=1}^{M} \sum_{i \neq j}^{n} L_h'\big(\epsilon_i^{(k)} - \epsilon_j^{(k)}\big) \big(\boldsymbol{X}_i^{(k)} - \boldsymbol{X}_j^{(k)}\big) \Big\|_\infty$$

$$= O_p\Big(M^{1/2} \Big\{ \big[s \log^{(5-4q)/(2-2q)} p / \sqrt{n}\big]^{1-q} + s^2 \log^{5/2} p / \sqrt{n} \Big\}\Big).$$

$\square$

*Proof of Theorem 3.2.* The proof is similar to that of Theorem 2.2. We omit it.  $\square$

*Proof of Theorem 3.3.* According to Theorem 3.2, for all $t \in \mathbb{R}$ we have

$$\big| \mathbb{P}\big(\sqrt{N} \, \|\widetilde{\boldsymbol{\beta}}_{\mathrm{ode}} - \boldsymbol{\beta}^*\|_\infty \geq t\big) - \mathbb{P}\big(\|\boldsymbol{Z}_n\|_\infty \geq t\big) \big| = o(1).$$

Then repeat the proof of Theorem 2.3.  $\square$

*Proof of Theorem 3.4.* Theorem 2.3 and 3.3 show that

$$\|\widetilde{\boldsymbol{\beta}} - \boldsymbol{\beta}^*\|_\infty = O_p\Big(\sqrt{\frac{\log p}{N}}\Big),$$

where $\widetilde{\boldsymbol{\beta}}$ is ADE or ODE. So there exists a constant $b_0 > 0$ satisfies $\|\widetilde{\boldsymbol{\beta}} - \boldsymbol{\beta}^*\|_\infty \leq b_0\sqrt{\log p/N}$ with high probability. Choose $c_0 > b_0$, then the support of $\widetilde{\boldsymbol{\beta}}^c$ is contained in the support of $\boldsymbol{\beta}^*$ with high probability. Recall that $\mathcal{S} = \{j : \beta_j^* \neq 0\}$ and $s = |\mathcal{S}|$. Let $c = c_0\sqrt{\log p/N}$. By a direct calculation, we have that

$$\|\widetilde{\boldsymbol{\beta}}^c - \boldsymbol{\beta}^*\|_\infty \leq \|\widetilde{\boldsymbol{\beta}}^c - \widetilde{\boldsymbol{\beta}}\|_\infty + \|\widetilde{\boldsymbol{\beta}} - \boldsymbol{\beta}^*\|_\infty = O_p\Big(c + \sqrt{\tfrac{\log p}{N}}\Big) = O_p\Big(\sqrt{\tfrac{\log p}{N}}\Big),$$

$$\|\widetilde{\boldsymbol{\beta}}^c - \boldsymbol{\beta}^*\|_2 = \Big(\sum_{k \in \mathcal{S}}(\widetilde{\beta}_k^c - \beta_k^*)^2\Big)^{1/2} \leq \sqrt{s}\,\|\widetilde{\boldsymbol{\beta}}^c - \boldsymbol{\beta}^*\|_\infty = O_p\Big(\sqrt{\tfrac{s\log p}{N}}\Big),$$

$$\|\widetilde{\boldsymbol{\beta}}^c - \boldsymbol{\beta}^*\|_1 = \sum_{k \in \mathcal{S}}|\widetilde{\beta}_k^c - \beta_k^*| \leq s\,\|\widetilde{\boldsymbol{\beta}}^c - \boldsymbol{\beta}^*\|_\infty = O_p\Big(s\sqrt{\tfrac{\log p}{N}}\Big).$$

$\square$

*Proof of Corollary 3.1.* Under the stated assumptions, the proofs of Theorem 2.3 and Theorem 3.2 yield that there exists a sufficiently large constant $b_0 > 0$ such that

$$\mathbb{P}\Big(\|\widetilde{\boldsymbol{\beta}} - \boldsymbol{\beta}^*\|_\infty \leq b_0\sqrt{\log p/N}\Big) \to 1.$$

Choose $c_0 > b_0$ and $c = c_0\sqrt{\log p/N}$.

For any $j \notin \mathcal{S}$, $\beta_j^* = 0$, and on the event $\{\|\widetilde{\boldsymbol{\beta}} - \boldsymbol{\beta}^*\|_\infty \leq b_0\sqrt{\log p/N}\}$ we have $|\widetilde{\beta}_j| \leq b_0\sqrt{\log p/N} < c$, hence $\widetilde{\beta}_j^c = 0$. Therefore $\mathrm{supp}(\widetilde{\boldsymbol{\beta}}^c) \subseteq \mathcal{S}$ with probability tending to one.

Assume $\beta_{\min} \geq 2c$. For any $j \in \mathcal{S}$, on the same event,

$$|\widetilde{\beta}_j| \geq |\beta_j^*| - |\widetilde{\beta}_j - \beta_j^*| \geq 2c - b_0\sqrt{\log p/N} > c,$$

since $c_0 > b_0$. Hence $\widetilde{\beta}_j^c = \widetilde{\beta}_j \neq 0$ for all $j \in \mathcal{S}$, implying $\mathcal{S} \subseteq \mathrm{supp}(\widetilde{\boldsymbol{\beta}}^c)$. Combining with part (i) yields $\mathrm{supp}(\widetilde{\boldsymbol{\beta}}^c) = \mathcal{S}$ with probability tending to one. $\square$

# B  Additional simulations

We consider sample size in each local machine to be $n = 200$ and $p = 100, 200$. The results are calculated based on 500 simulation runs.

## B.1  Different choices of bandwidths $h$

We also consider different bandwidths $h = 0.1, 0.5, 1.5$. The corresponding results are shown in Table S.1, which indicates the hyperparameter $h$ can be tuned over a wide range.

## B.2  Comparisons with existing methods

Table S.1: Empirical average coverage probability (ACP) and average lengths (AL) of confidence intervals for ADE and ODE at the confidence level 95% under **Scenario 1** and **Setting 1** with $p = 100$.

| | | | $\mathcal{G}_1$ | | $\mathcal{G}_2$ | | $\mathcal{G}_3$ | |
| --- | --- | --- | --- | --- | --- | --- | --- | --- |
| | | $M$ | ACP | AL | ACP | AL | ACP | AL |
| $h=0.1$ | ADE | 5 | 0.946 | 0.136 | 0.948 | 0.136 | 0.947 | 0.136 |
| | | 10 | 0.943 | 0.096 | 0.945 | 0.096 | 0.944 | 0.096 |
| | | 15 | 0.945 | 0.078 | 0.945 | 0.078 | 0.945 | 0.078 |
| | | 20 | 0.945 | 0.068 | 0.944 | 0.068 | 0.944 | 0.068 |
| | ODE | 5 | 0.937 | 0.136 | 0.937 | 0.136 | 0.937 | 0.136 |
| | | 10 | 0.929 | 0.096 | 0.936 | 0.096 | 0.933 | 0.096 |
| | | 15 | 0.928 | 0.079 | 0.931 | 0.079 | 0.929 | 0.079 |
| | | 20 | 0.927 | 0.068 | 0.933 | 0.068 | 0.930 | 0.068 |
| $h=0.5$ | ADE | 5 | 0.947 | 0.137 | 0.948 | 0.137 | 0.947 | 0.137 |
| | | 10 | 0.943 | 0.097 | 0.946 | 0.097 | 0.944 | 0.097 |
| | | 15 | 0.945 | 0.079 | 0.945 | 0.079 | 0.945 | 0.079 |
| | | 20 | 0.945 | 0.069 | 0.943 | 0.069 | 0.944 | 0.069 |
| | ODE | 5 | 0.936 | 0.137 | 0.937 | 0.137 | 0.937 | 0.137 |
| | | 10 | 0.929 | 0.097 | 0.936 | 0.097 | 0.933 | 0.097 |
| | | 15 | 0.928 | 0.079 | 0.931 | 0.079 | 0.930 | 0.079 |
| | | 20 | 0.929 | 0.069 | 0.932 | 0.069 | 0.930 | 0.069 |
| $h=1.5$ | ADE | 5 | 0.948 | 0.151 | 0.948 | 0.151 | 0.948 | 0.151 |
| | | 10 | 0.944 | 0.107 | 0.946 | 0.107 | 0.945 | 0.107 |
| | | 15 | 0.946 | 0.087 | 0.945 | 0.087 | 0.945 | 0.087 |
| | | 20 | 0.946 | 0.075 | 0.943 | 0.075 | 0.945 | 0.075 |
| | ODE | 5 | 0.938 | 0.151 | 0.937 | 0.151 | 0.937 | 0.151 |
| | | 10 | 0.929 | 0.107 | 0.936 | 0.106 | 0.933 | 0.106 |
| | | 15 | 0.928 | 0.087 | 0.931 | 0.087 | 0.930 | 0.087 |
| | | 20 | 0.928 | 0.075 | 0.932 | 0.075 | 0.930 | 0.075 |

For comparison, we also implement the CSSDS estimation procedure proposed by Di et al. (2022), which is designed for inference on low-dimensional parameters in the presence of high-dimensional nuisance covariates. We then focus on inference for the coefficients indexed by $\mathcal{G}_1^* = \{1\}$ and $\mathcal{G}_2^* = \{1, 2, 3, 4, 5\}$. The corresponding results are shown in Tables S.2-S.5. Compared with CSSDS1 and CSSDS2, the coverage probabilities of ADE and ODE are closer to the nominal level of 95%. Moreover, the average lengths of the

confidence intervals for ADE and ODE are shorter than those for CSSDS1 and CSSDS2.

Table S.2: Empirical average coverage probability (ACP) and average lengths (AL) of confidence intervals at the confidence level 95% under **Scenario 1** and **Setting 1**.

| | | $M$ | ADE ACP | ADE AL | ODE ACP | ODE AL | CSSDS1 ACP | CSSDS1 AL | CSSDS2 ACP | CSSDS2 AL |
|---|---|---|---|---|---|---|---|---|---|---|
| $p = 100$ | $\mathcal{G}_1^*$ | 5 | 0.950 | 0.137 | 0.910 | 0.137 | 0.920 | 0.165 | 0.910 | 0.165 |
| | | 10 | 0.955 | 0.097 | 0.910 | 0.097 | 0.905 | 0.117 | 0.910 | 0.117 |
| | | 15 | 0.945 | 0.079 | 0.915 | 0.079 | 0.875 | 0.096 | 0.855 | 0.096 |
| | | 20 | 0.965 | 0.069 | 0.900 | 0.068 | 0.785 | 0.083 | 0.785 | 0.083 |
| | $\mathcal{G}_2^*$ | 5 | 0.957 | 0.141 | 0.942 | 0.141 | 0.984 | 0.178 | 0.982 | 0.178 |
| | | 10 | 0.963 | 0.100 | 0.923 | 0.100 | 0.982 | 0.127 | 0.983 | 0.127 |
| | | 15 | 0.953 | 0.082 | 0.900 | 0.082 | 0.978 | 0.104 | 0.975 | 0.103 |
| | | 20 | 0.955 | 0.071 | 0.900 | 0.071 | 0.979 | 0.090 | 0.977 | 0.090 |
| $p = 200$ | $\mathcal{G}_1^*$ | 5 | 0.975 | 0.140 | 0.940 | 0.139 | 0.925 | 0.165 | 0.930 | 0.165 |
| | | 10 | 0.970 | 0.099 | 0.900 | 0.099 | 0.860 | 0.117 | 0.840 | 0.117 |
| | | 15 | 0.950 | 0.081 | 0.900 | 0.080 | 0.800 | 0.096 | 0.770 | 0.096 |
| | | 20 | 0.945 | 0.070 | 0.900 | 0.070 | 0.720 | 0.083 | 0.710 | 0.083 |
| | $\mathcal{G}_2^*$ | 5 | 0.950 | 0.144 | 0.921 | 0.143 | 0.976 | 0.177 | 0.981 | 0.177 |
| | | 10 | 0.948 | 0.102 | 0.900 | 0.101 | 0.970 | 0.126 | 0.977 | 0.126 |
| | | 15 | 0.947 | 0.083 | 0.900 | 0.083 | 0.975 | 0.103 | 0.977 | 0.103 |
| | | 20 | 0.948 | 0.072 | 0.900 | 0.072 | 0.977 | 0.089 | 0.976 | 0.089 |

## B.3 Different choices of error distributions

We consider other error distributions. The errors $\epsilon_i$'s are generated from the following distributions:

- **Scenario 1':** $\epsilon_i$ is simulated from a $t$-distribution with 3 degrees of freedom.

- **Scenario 2':** $\epsilon_i$ follows the standard Cauchy distribution.

- **Scenario 3':** $\epsilon_i$ follows a skew-normal distribution with shape parameter 0.5 (positively skewed).

- **Scenario 4':** $\epsilon_i$ follows a skew-normal distribution with shape parameter –0.5 (negatively skewed).

Table S.3: Empirical average coverage probability (ACP) and average lengths (AL) of confidence intervals at the confidence level 95% under **Scenario 1** and **Setting 2**.

|  |  | $M$ | ADE ACP | ADE AL | ODE ACP | ODE AL | CSSDS1 ACP | CSSDS1 AL | CSSDS2 ACP | CSSDS2 AL |
|---|---|---|---|---|---|---|---|---|---|---|
| $p = 100$ | $\mathcal{G}_1^*$ | 5 | 0.935 | 0.137 | 0.905 | 0.137 | 0.910 | 0.165 | 0.915 | 0.165 |
|  |  | 10 | 0.935 | 0.097 | 0.920 | 0.097 | 0.945 | 0.117 | 0.940 | 0.117 |
|  |  | 15 | 0.935 | 0.079 | 0.900 | 0.079 | 0.915 | 0.096 | 0.915 | 0.096 |
|  |  | 20 | 0.945 | 0.069 | 0.900 | 0.068 | 0.920 | 0.083 | 0.905 | 0.083 |
|  | $\mathcal{G}_2^*$ | 5 | 0.956 | 0.142 | 0.926 | 0.142 | 0.977 | 0.181 | 0.981 | 0.181 |
|  |  | 10 | 0.944 | 0.101 | 0.916 | 0.100 | 0.972 | 0.128 | 0.970 | 0.128 |
|  |  | 15 | 0.944 | 0.082 | 0.900 | 0.082 | 0.966 | 0.105 | 0.971 | 0.105 |
|  |  | 20 | 0.950 | 0.071 | 0.900 | 0.071 | 0.975 | 0.091 | 0.976 | 0.091 |
| $p = 200$ | $\mathcal{G}_1^*$ | 5 | 0.965 | 0.140 | 0.930 | 0.140 | 0.955 | 0.166 | 0.965 | 0.165 |
|  |  | 10 | 0.975 | 0.099 | 0.900 | 0.099 | 0.950 | 0.117 | 0.945 | 0.117 |
|  |  | 15 | 0.950 | 0.081 | 0.900 | 0.081 | 0.925 | 0.096 | 0.930 | 0.096 |
|  |  | 20 | 0.975 | 0.070 | 0.900 | 0.070 | 0.910 | 0.083 | 0.915 | 0.083 |
|  | $\mathcal{G}_2^*$ | 5 | 0.963 | 0.145 | 0.938 | 0.144 | 0.986 | 0.179 | 0.985 | 0.179 |
|  |  | 10 | 0.954 | 0.103 | 0.900 | 0.102 | 0.982 | 0.127 | 0.982 | 0.127 |
|  |  | 15 | 0.952 | 0.084 | 0.900 | 0.083 | 0.976 | 0.104 | 0.983 | 0.104 |
|  |  | 20 | 0.964 | 0.073 | 0.900 | 0.072 | 0.983 | 0.091 | 0.985 | 0.091 |

Tables S.6-S.13 present the results. Our method maintains coverage close to 95% in all cases. The intervals are also consistently short, even when the errors follow a t-distribution, the Cauchy distribution, or asymmetric distributions.

## B.4  Different choices of $\boldsymbol{\beta}$

To evaluate the performance under different true coefficients, we additionally consider $\boldsymbol{\beta}^* = (2, 1.5, 1, 0, \ldots, 0)^\top$. The numerical results presented in Tables S.14-S.17 suggest that the choice of true coefficients has little effect on the coverage and accuracy of our proposed method.

## B.5  Different choices of threshold $c$

We consider the heuristic rule of threshold $c$. Specifically, let $s_{\max} = \max_{1 \leq k \leq M} \|\widehat{\boldsymbol{\beta}}_{kh}\|_0$, order $|\widetilde{\beta}|_{(1)} \geq \cdots \geq |\widetilde{\beta}|_{(p)}$, and set $c = |\widetilde{\beta}|_{(s_{\max}+1)}$. Here $|\widetilde{\beta}|_{(l)}$ represents the $l$-th largest element of the absolute value version of the ADE or the ODE. The numerical results are

Table S.4: Empirical average coverage probability (ACP) and average lengths (AL) of confidence intervals at the confidence level 95% under **Scenario 2** and **Setting 1**.

| | | $M$ | ADE ACP | ADE AL | ODE ACP | ODE AL | CSSDS1 ACP | CSSDS1 AL | CSSDS2 ACP | CSSDS2 AL |
|---|---|---|---|---|---|---|---|---|---|---|
| $p = 100$ | $\mathcal{G}_1^*$ | 5 | 0.935 | 0.145 | 0.915 | 0.145 | 0.945 | 0.172 | 0.945 | 0.171 |
| | | 10 | 0.935 | 0.103 | 0.900 | 0.102 | 0.875 | 0.122 | 0.885 | 0.121 |
| | | 15 | 0.975 | 0.084 | 0.900 | 0.084 | 0.855 | 0.100 | 0.865 | 0.099 |
| | | 20 | 0.965 | 0.073 | 0.900 | 0.072 | 0.820 | 0.086 | 0.825 | 0.086 |
| | $\mathcal{G}_2^*$ | 5 | 0.949 | 0.150 | 0.925 | 0.150 | 0.973 | 0.185 | 0.973 | 0.185 |
| | | 10 | 0.943 | 0.106 | 0.917 | 0.106 | 0.972 | 0.131 | 0.974 | 0.131 |
| | | 15 | 0.953 | 0.087 | 0.900 | 0.086 | 0.980 | 0.107 | 0.979 | 0.107 |
| | | 20 | 0.954 | 0.075 | 0.900 | 0.075 | 0.980 | 0.093 | 0.978 | 0.093 |
| $p = 200$ | $\mathcal{G}_1^*$ | 5 | 0.960 | 0.149 | 0.920 | 0.148 | 0.900 | 0.172 | 0.885 | 0.172 |
| | | 10 | 0.960 | 0.105 | 0.910 | 0.105 | 0.885 | 0.122 | 0.870 | 0.122 |
| | | 15 | 0.975 | 0.086 | 0.900 | 0.085 | 0.770 | 0.100 | 0.765 | 0.100 |
| | | 20 | 0.980 | 0.074 | 0.900 | 0.074 | 0.730 | 0.086 | 0.730 | 0.086 |
| | $\mathcal{G}_2^*$ | 5 | 0.954 | 0.153 | 0.925 | 0.152 | 0.981 | 0.184 | 0.979 | 0.184 |
| | | 10 | 0.957 | 0.108 | 0.906 | 0.107 | 0.980 | 0.131 | 0.980 | 0.131 |
| | | 15 | 0.958 | 0.088 | 0.900 | 0.088 | 0.983 | 0.107 | 0.984 | 0.107 |
| | | 20 | 0.965 | 0.077 | 0.900 | 0.076 | 0.983 | 0.093 | 0.987 | 0.093 |

presented in Figures S.1-S.3. The results under this heuristic rule are similar to those obtained using the approach $c = c_0\sqrt{\log(p)/N}$ with $c_0 = 1.5$. The estimation errors decrease as the sample size $N$ grows, but they increase with the number of machines $M$ and the dimensionality $p$. For the $\ell_2$ error, the sparse aggregated estimators perform better than their non-sparse counterparts. For the $\ell_\infty$ error, the differences among methods are modest.

Table S.5: Empirical average coverage probability (ACP) and average lengths (AL) of confidence intervals at the confidence level 95% under **Scenario 2** and **Setting 2**.

| | | $M$ | ADE ACP | ADE AL | ODE ACP | ODE AL | CSSDS1 ACP | CSSDS1 AL | CSSDS2 ACP | CSSDS2 AL |
|---|---|---|---|---|---|---|---|---|---|---|
| $p=100$ | $\mathcal{G}_1^*$ | 5 | 0.940 | 0.145 | 0.920 | 0.145 | 0.955 | 0.172 | 0.950 | 0.171 |
| | | 10 | 0.960 | 0.103 | 0.900 | 0.102 | 0.965 | 0.122 | 0.975 | 0.122 |
| | | 15 | 0.945 | 0.084 | 0.900 | 0.084 | 0.950 | 0.100 | 0.945 | 0.099 |
| | | 20 | 0.925 | 0.073 | 0.900 | 0.072 | 0.930 | 0.086 | 0.925 | 0.086 |
| | $\mathcal{G}_2^*$ | 5 | 0.947 | 0.151 | 0.925 | 0.150 | 0.970 | 0.187 | 0.974 | 0.187 |
| | | 10 | 0.956 | 0.107 | 0.906 | 0.106 | 0.979 | 0.133 | 0.981 | 0.133 |
| | | 15 | 0.952 | 0.087 | 0.900 | 0.087 | 0.981 | 0.109 | 0.986 | 0.109 |
| | | 20 | 0.948 | 0.076 | 0.900 | 0.075 | 0.982 | 0.094 | 0.983 | 0.094 |
| $p=200$ | $\mathcal{G}_1^*$ | 5 | 0.980 | 0.149 | 0.920 | 0.148 | 0.930 | 0.172 | 0.940 | 0.172 |
| | | 10 | 0.975 | 0.105 | 0.910 | 0.105 | 0.920 | 0.122 | 0.925 | 0.122 |
| | | 15 | 0.980 | 0.086 | 0.900 | 0.085 | 0.910 | 0.100 | 0.905 | 0.099 |
| | | 20 | 0.960 | 0.074 | 0.900 | 0.074 | 0.885 | 0.086 | 0.890 | 0.086 |
| | $\mathcal{G}_2^*$ | 5 | 0.957 | 0.154 | 0.928 | 0.153 | 0.982 | 0.186 | 0.984 | 0.186 |
| | | 10 | 0.958 | 0.109 | 0.900 | 0.108 | 0.979 | 0.133 | 0.981 | 0.133 |
| | | 15 | 0.956 | 0.089 | 0.900 | 0.088 | 0.982 | 0.109 | 0.982 | 0.108 |
| | | 20 | 0.941 | 0.077 | 0.900 | 0.076 | 0.975 | 0.094 | 0.981 | 0.094 |

Table S.6: Empirical average coverage probability (ACP) and average lengths (AL) of confidence intervals for ADE and ODE at the confidence level 95% under **Scenario 1'** with $p = 100$.

|  |  | $M$ | $\mathcal{G}_1$ ACP | $\mathcal{G}_1$ AL | $\mathcal{G}_2$ ACP | $\mathcal{G}_2$ AL | $\mathcal{G}_3$ ACP | $\mathcal{G}_3$ AL |
|---|---|---|---|---|---|---|---|---|
| **Setting 1** | ADE | 5 | 0.945 | 0.177 | 0.948 | 0.177 | 0.947 | 0.177 |
|  |  | 10 | 0.945 | 0.125 | 0.945 | 0.125 | 0.945 | 0.125 |
|  |  | 15 | 0.946 | 0.102 | 0.944 | 0.102 | 0.945 | 0.102 |
|  |  | 20 | 0.943 | 0.089 | 0.943 | 0.089 | 0.943 | 0.089 |
|  | ODE | 5 | 0.935 | 0.178 | 0.937 | 0.178 | 0.936 | 0.178 |
|  |  | 10 | 0.928 | 0.126 | 0.936 | 0.126 | 0.932 | 0.126 |
|  |  | 15 | 0.926 | 0.103 | 0.932 | 0.103 | 0.929 | 0.103 |
|  |  | 20 | 0.922 | 0.089 | 0.931 | 0.089 | 0.927 | 0.089 |
| **Setting 2** | ADE | 5 | 0.943 | 0.179 | 0.945 | 0.179 | 0.944 | 0.179 |
|  |  | 10 | 0.944 | 0.127 | 0.945 | 0.127 | 0.945 | 0.127 |
|  |  | 15 | 0.945 | 0.103 | 0.943 | 0.103 | 0.944 | 0.103 |
|  |  | 20 | 0.945 | 0.090 | 0.940 | 0.090 | 0.942 | 0.090 |
|  | ODE | 5 | 0.934 | 0.179 | 0.932 | 0.179 | 0.933 | 0.179 |
|  |  | 10 | 0.931 | 0.127 | 0.932 | 0.127 | 0.932 | 0.127 |
|  |  | 15 | 0.929 | 0.103 | 0.932 | 0.104 | 0.930 | 0.104 |
|  |  | 20 | 0.923 | 0.090 | 0.928 | 0.090 | 0.925 | 0.090 |

Table S.7: Empirical average coverage probability (ACP) and average lengths (AL) of confidence intervals for ADE and ODE at the confidence level 95% under **Scenario 2'** with $p = 100$.

| | | | $\mathcal{G}_1$ | | $\mathcal{G}_2$ | | $\mathcal{G}_3$ | |
| --- | --- | --- | --- | --- | --- | --- | --- | --- |
| | | $M$ | ACP | AL | ACP | AL | ACP | AL |
| **Setting 1** | ADE | 5 | 0.947 | 0.266 | 0.946 | 0.266 | 0.947 | 0.266 |
| | | 10 | 0.944 | 0.188 | 0.947 | 0.188 | 0.945 | 0.188 |
| | | 15 | 0.945 | 0.153 | 0.947 | 0.153 | 0.946 | 0.153 |
| | | 20 | 0.944 | 0.133 | 0.943 | 0.133 | 0.944 | 0.133 |
| | ODE | 5 | 0.935 | 0.266 | 0.938 | 0.266 | 0.937 | 0.266 |
| | | 10 | 0.930 | 0.188 | 0.935 | 0.188 | 0.932 | 0.188 |
| | | 15 | 0.928 | 0.154 | 0.933 | 0.153 | 0.931 | 0.153 |
| | | 20 | 0.923 | 0.133 | 0.931 | 0.133 | 0.927 | 0.133 |
| **Setting 2** | ADE | 5 | 0.944 | 0.269 | 0.947 | 0.269 | 0.945 | 0.269 |
| | | 10 | 0.946 | 0.190 | 0.946 | 0.190 | 0.946 | 0.190 |
| | | 15 | 0.945 | 0.155 | 0.944 | 0.155 | 0.945 | 0.155 |
| | | 20 | 0.945 | 0.134 | 0.947 | 0.134 | 0.946 | 0.134 |
| | ODE | 5 | 0.932 | 0.269 | 0.936 | 0.269 | 0.934 | 0.269 |
| | | 10 | 0.929 | 0.190 | 0.935 | 0.190 | 0.932 | 0.190 |
| | | 15 | 0.925 | 0.155 | 0.933 | 0.155 | 0.929 | 0.155 |
| | | 20 | 0.921 | 0.134 | 0.936 | 0.134 | 0.928 | 0.134 |

Table S.8: Empirical average coverage probability (ACP) and average lengths (AL) of confidence intervals for ADE and ODE at the confidence level 95% under **Scenario 3'** with $p = 100$.

| | | $M$ | $\mathcal{G}_1$ ACP | $\mathcal{G}_1$ AL | $\mathcal{G}_2$ ACP | $\mathcal{G}_2$ AL | $\mathcal{G}_3$ ACP | $\mathcal{G}_3$ AL |
|---|---|---|---|---|---|---|---|---|
| **Setting 1** | ADE | 5 | 0.947 | 0.133 | 0.944 | 0.133 | 0.945 | 0.133 |
| | | 10 | 0.951 | 0.094 | 0.946 | 0.094 | 0.948 | 0.094 |
| | | 15 | 0.947 | 0.077 | 0.942 | 0.077 | 0.945 | 0.077 |
| | | 20 | 0.947 | 0.066 | 0.944 | 0.066 | 0.946 | 0.066 |
| | ODE | 5 | 0.936 | 0.133 | 0.936 | 0.133 | 0.936 | 0.133 |
| | | 10 | 0.937 | 0.094 | 0.937 | 0.094 | 0.936 | 0.094 |
| | | 15 | 0.931 | 0.077 | 0.931 | 0.077 | 0.932 | 0.077 |
| | | 20 | 0.925 | 0.067 | 0.925 | 0.067 | 0.930 | 0.067 |
| **Setting 2** | ADE | 5 | 0.946 | 0.134 | 0.944 | 0.134 | 0.945 | 0.134 |
| | | 10 | 0.945 | 0.095 | 0.944 | 0.095 | 0.944 | 0.095 |
| | | 15 | 0.948 | 0.077 | 0.945 | 0.077 | 0.946 | 0.077 |
| | | 20 | 0.947 | 0.067 | 0.945 | 0.067 | 0.946 | 0.067 |
| | ODE | 5 | 0.935 | 0.134 | 0.931 | 0.135 | 0.933 | 0.135 |
| | | 10 | 0.930 | 0.095 | 0.931 | 0.095 | 0.931 | 0.095 |
| | | 15 | 0.932 | 0.078 | 0.932 | 0.078 | 0.932 | 0.078 |
| | | 20 | 0.928 | 0.067 | 0.934 | 0.067 | 0.931 | 0.067 |

Table S.9: Empirical average coverage probability (ACP) and average lengths (AL) of confidence intervals for ADE and ODE at the confidence level 95% under **Scenario 4'** with $p = 100$.

| | | | $\mathcal{G}_1$ | | $\mathcal{G}_2$ | | $\mathcal{G}_3$ | |
| --- | --- | --- | --- | --- | --- | --- | --- | --- |
| | | $M$ | ACP | AL | ACP | AL | ACP | AL |
| Setting 1 | ADE | 5 | 0.950 | 0.133 | 0.943 | 0.133 | 0.947 | 0.133 |
| | | 10 | 0.950 | 0.094 | 0.943 | 0.094 | 0.947 | 0.094 |
| | | 15 | 0.953 | 0.077 | 0.944 | 0.077 | 0.948 | 0.077 |
| | | 20 | 0.950 | 0.066 | 0.942 | 0.066 | 0.946 | 0.066 |
| | ODE | 5 | 0.939 | 0.133 | 0.936 | 0.133 | 0.938 | 0.133 |
| | | 10 | 0.933 | 0.094 | 0.934 | 0.094 | 0.933 | 0.094 |
| | | 15 | 0.934 | 0.077 | 0.929 | 0.077 | 0.932 | 0.077 |
| | | 20 | 0.931 | 0.067 | 0.931 | 0.066 | 0.931 | 0.067 |
| Setting 2 | ADE | 5 | 0.944 | 0.134 | 0.942 | 0.134 | 0.943 | 0.134 |
| | | 10 | 0.945 | 0.095 | 0.946 | 0.095 | 0.945 | 0.095 |
| | | 15 | 0.948 | 0.077 | 0.945 | 0.077 | 0.947 | 0.077 |
| | | 20 | 0.946 | 0.067 | 0.947 | 0.067 | 0.946 | 0.067 |
| | ODE | 5 | 0.935 | 0.134 | 0.935 | 0.134 | 0.935 | 0.134 |
| | | 10 | 0.931 | 0.095 | 0.935 | 0.095 | 0.933 | 0.095 |
| | | 15 | 0.933 | 0.078 | 0.933 | 0.078 | 0.933 | 0.078 |
| | | 20 | 0.928 | 0.067 | 0.931 | 0.067 | 0.929 | 0.067 |

Table S.10: Empirical average coverage probability (ACP) and average lengths (AL) of confidence intervals for ADE and ODE at the confidence level 95% under **Scenario 1'** with $p = 200$.

| | | | $\mathcal{G}_1$ | | $\mathcal{G}_2$ | | $\mathcal{G}_3$ | |
|---|---|---|---|---|---|---|---|---|
| | | $M$ | ACP | AL | ACP | AL | ACP | AL |
| **Setting 1** | ADE | 5 | 0.944 | 0.180 | 0.942 | 0.180 | 0.943 | 0.180 |
| | | 10 | 0.942 | 0.128 | 0.943 | 0.128 | 0.942 | 0.128 |
| | | 15 | 0.945 | 0.104 | 0.944 | 0.104 | 0.944 | 0.104 |
| | | 20 | 0.942 | 0.090 | 0.943 | 0.090 | 0.942 | 0.090 |
| | ODE | 5 | 0.929 | 0.179 | 0.929 | 0.179 | 0.929 | 0.179 |
| | | 10 | 0.927 | 0.127 | 0.929 | 0.127 | 0.928 | 0.127 |
| | | 15 | 0.925 | 0.104 | 0.926 | 0.104 | 0.926 | 0.104 |
| | | 20 | 0.920 | 0.090 | 0.925 | 0.090 | 0.922 | 0.090 |
| **Setting 2** | ADE | 5 | 0.943 | 0.182 | 0.942 | 0.182 | 0.942 | 0.182 |
| | | 10 | 0.942 | 0.129 | 0.945 | 0.129 | 0.944 | 0.129 |
| | | 15 | 0.943 | 0.105 | 0.944 | 0.105 | 0.943 | 0.105 |
| | | 20 | 0.944 | 0.091 | 0.943 | 0.091 | 0.944 | 0.091 |
| | ODE | 5 | 0.928 | 0.181 | 0.929 | 0.181 | 0.928 | 0.181 |
| | | 10 | 0.924 | 0.128 | 0.928 | 0.128 | 0.926 | 0.128 |
| | | 15 | 0.921 | 0.104 | 0.929 | 0.104 | 0.925 | 0.104 |
| | | 20 | 0.920 | 0.090 | 0.927 | 0.090 | 0.923 | 0.090 |

Table S.11: Empirical average coverage probability (ACP) and average lengths (AL) of confidence intervals for ADE and ODE at the confidence level 95% under **Scenario 2'** with $p = 200$.

| | | $M$ | $\mathcal{G}_1$ ACP | $\mathcal{G}_1$ AL | $\mathcal{G}_2$ ACP | $\mathcal{G}_2$ AL | $\mathcal{G}_3$ ACP | $\mathcal{G}_3$ AL |
|---|---|---|---|---|---|---|---|---|
| **Setting 1** | ADE | 5 | 0.943 | 0.274 | 0.946 | 0.274 | 0.945 | 0.274 |
| | | 10 | 0.942 | 0.194 | 0.944 | 0.194 | 0.943 | 0.194 |
| | | 15 | 0.943 | 0.158 | 0.944 | 0.158 | 0.944 | 0.158 |
| | | 20 | 0.941 | 0.137 | 0.946 | 0.137 | 0.944 | 0.137 |
| | ODE | 5 | 0.930 | 0.273 | 0.933 | 0.273 | 0.932 | 0.273 |
| | | 10 | 0.924 | 0.193 | 0.927 | 0.193 | 0.925 | 0.193 |
| | | 15 | 0.922 | 0.158 | 0.925 | 0.158 | 0.924 | 0.158 |
| | | 20 | 0.918 | 0.137 | 0.926 | 0.137 | 0.922 | 0.137 |
| **Setting 2** | ADE | 5 | 0.945 | 0.276 | 0.947 | 0.276 | 0.946 | 0.276 |
| | | 10 | 0.945 | 0.195 | 0.942 | 0.195 | 0.943 | 0.195 |
| | | 15 | 0.945 | 0.159 | 0.942 | 0.159 | 0.943 | 0.159 |
| | | 20 | 0.942 | 0.138 | 0.944 | 0.138 | 0.943 | 0.138 |
| | ODE | 5 | 0.929 | 0.276 | 0.933 | 0.276 | 0.931 | 0.276 |
| | | 10 | 0.925 | 0.195 | 0.927 | 0.195 | 0.926 | 0.195 |
| | | 15 | 0.924 | 0.159 | 0.925 | 0.159 | 0.924 | 0.159 |
| | | 20 | 0.918 | 0.138 | 0.925 | 0.138 | 0.922 | 0.138 |

Table S.12: Empirical average coverage probability (ACP) and average lengths (AL) of confidence intervals for ADE and ODE at the confidence level 95% under **Scenario 3'** with $p = 200$.

| | | | $\mathcal{G}_1$ | | $\mathcal{G}_2$ | | $\mathcal{G}_3$ | |
| --- | --- | --- | --- | --- | --- | --- | --- | --- |
| | | $M$ | ACP | AL | ACP | AL | ACP | AL |
| **Setting 1** | ADE | 5 | 0.942 | 0.135 | 0.944 | 0.135 | 0.943 | 0.135 |
| | | 10 | 0.937 | 0.095 | 0.943 | 0.095 | 0.940 | 0.095 |
| | | 15 | 0.941 | 0.078 | 0.944 | 0.078 | 0.943 | 0.078 |
| | | 20 | 0.942 | 0.067 | 0.944 | 0.067 | 0.943 | 0.067 |
| | ODE | 5 | 0.927 | 0.133 | 0.929 | 0.133 | 0.928 | 0.133 |
| | | 10 | 0.921 | 0.094 | 0.928 | 0.094 | 0.925 | 0.094 |
| | | 15 | 0.920 | 0.077 | 0.925 | 0.077 | 0.923 | 0.077 |
| | | 20 | 0.920 | 0.067 | 0.926 | 0.067 | 0.923 | 0.067 |
| **Setting 2** | ADE | 5 | 0.944 | 0.136 | 0.942 | 0.136 | 0.943 | 0.136 |
| | | 10 | 0.940 | 0.096 | 0.941 | 0.096 | 0.941 | 0.096 |
| | | 15 | 0.942 | 0.078 | 0.942 | 0.078 | 0.942 | 0.078 |
| | | 20 | 0.942 | 0.068 | 0.945 | 0.068 | 0.943 | 0.068 |
| | ODE | 5 | 0.928 | 0.135 | 0.927 | 0.135 | 0.927 | 0.135 |
| | | 10 | 0.922 | 0.095 | 0.927 | 0.095 | 0.924 | 0.095 |
| | | 15 | 0.921 | 0.078 | 0.925 | 0.078 | 0.923 | 0.078 |
| | | 20 | 0.922 | 0.067 | 0.928 | 0.067 | 0.925 | 0.067 |

Table S.13: Empirical average coverage probability (ACP) and average lengths (AL) of confidence intervals for ADE and ODE at the confidence level 95% under **Scenario 4'** with $p = 200$.

| | | | $\mathcal{G}_1$ | | $\mathcal{G}_2$ | | $\mathcal{G}_3$ | |
| --- | --- | --- | --- | --- | --- | --- | --- | --- |
| | | $M$ | ACP | AL | ACP | AL | ACP | AL |
| **Setting 1** | ADE | 5 | 0.944 | 0.135 | 0.943 | 0.135 | 0.943 | 0.135 |
| | | 10 | 0.942 | 0.096 | 0.943 | 0.096 | 0.942 | 0.096 |
| | | 15 | 0.941 | 0.078 | 0.945 | 0.078 | 0.943 | 0.078 |
| | | 20 | 0.938 | 0.068 | 0.944 | 0.067 | 0.941 | 0.068 |
| | ODE | 5 | 0.927 | 0.134 | 0.927 | 0.134 | 0.928 | 0.134 |
| | | 10 | 0.922 | 0.095 | 0.922 | 0.095 | 0.925 | 0.095 |
| | | 15 | 0.921 | 0.077 | 0.921 | 0.077 | 0.925 | 0.077 |
| | | 20 | 0.919 | 0.067 | 0.919 | 0.067 | 0.922 | 0.067 |
| **Setting 2** | ADE | 5 | 0.945 | 0.136 | 0.942 | 0.136 | 0.943 | 0.136 |
| | | 10 | 0.941 | 0.096 | 0.940 | 0.096 | 0.941 | 0.096 |
| | | 15 | 0.942 | 0.078 | 0.943 | 0.079 | 0.943 | 0.079 |
| | | 20 | 0.944 | 0.068 | 0.942 | 0.068 | 0.943 | 0.068 |
| | ODE | 5 | 0.931 | 0.135 | 0.928 | 0.135 | 0.929 | 0.135 |
| | | 10 | 0.922 | 0.096 | 0.925 | 0.096 | 0.924 | 0.096 |
| | | 15 | 0.923 | 0.078 | 0.926 | 0.078 | 0.925 | 0.078 |
| | | 20 | 0.921 | 0.068 | 0.925 | 0.068 | 0.923 | 0.068 |

Table S.14: Empirical average coverage probability (ACP) and average lengths (AL) of confidence intervals for ADE and ODE at the confidence level 95% under **Scenario 1** with $p = 100$.

| | | $M$ | $\mathcal{G}_1$ ACP | $\mathcal{G}_1$ AL | $\mathcal{G}_2$ ACP | $\mathcal{G}_2$ AL | $\mathcal{G}_3$ ACP | $\mathcal{G}_3$ AL |
|---|---|---|---|---|---|---|---|---|
| **Setting 1** | ADE | 5 | 0.948 | 0.142 | 0.946 | 0.142 | 0.947 | 0.142 |
| | | 10 | 0.943 | 0.101 | 0.946 | 0.101 | 0.944 | 0.101 |
| | | 15 | 0.944 | 0.082 | 0.946 | 0.082 | 0.945 | 0.082 |
| | | 20 | 0.945 | 0.071 | 0.944 | 0.071 | 0.944 | 0.071 |
| | ODE | 5 | 0.936 | 0.142 | 0.935 | 0.142 | 0.936 | 0.142 |
| | | 10 | 0.929 | 0.101 | 0.936 | 0.100 | 0.933 | 0.101 |
| | | 15 | 0.929 | 0.082 | 0.934 | 0.082 | 0.932 | 0.082 |
| | | 20 | 0.928 | 0.071 | 0.933 | 0.071 | 0.931 | 0.071 |
| **Setting 2** | ADE | 5 | 0.944 | 0.151 | 0.940 | 0.151 | 0.942 | 0.151 |
| | | 10 | 0.945 | 0.107 | 0.944 | 0.107 | 0.945 | 0.107 |
| | | 15 | 0.944 | 0.087 | 0.947 | 0.087 | 0.945 | 0.087 |
| | | 20 | 0.942 | 0.075 | 0.949 | 0.075 | 0.945 | 0.075 |
| | ODE | 5 | 0.932 | 0.151 | 0.931 | 0.151 | 0.931 | 0.151 |
| | | 10 | 0.930 | 0.107 | 0.934 | 0.106 | 0.932 | 0.107 |
| | | 15 | 0.927 | 0.087 | 0.936 | 0.087 | 0.932 | 0.087 |
| | | 20 | 0.926 | 0.075 | 0.935 | 0.075 | 0.930 | 0.075 |

Table S.15: Empirical average coverage probability (ACP) and average lengths (AL) of confidence intervals for ADE and ODE at the confidence level 95% under **Scenario 2** with $p = 100$.

| | | | $\mathcal{G}_1$ | | $\mathcal{G}_2$ | | $\mathcal{G}_3$ | |
|---|---|---|---|---|---|---|---|---|
| | | $M$ | ACP | AL | ACP | AL | ACP | AL |
| **Setting 1** | ADE | 5 | 0.944 | 0.151 | 0.940 | 0.151 | 0.942 | 0.151 |
| | | 10 | 0.945 | 0.107 | 0.944 | 0.107 | 0.945 | 0.107 |
| | | 15 | 0.944 | 0.087 | 0.947 | 0.087 | 0.945 | 0.087 |
| | | 20 | 0.942 | 0.075 | 0.949 | 0.075 | 0.945 | 0.075 |
| | ODE | 5 | 0.932 | 0.151 | 0.931 | 0.151 | 0.931 | 0.151 |
| | | 10 | 0.930 | 0.107 | 0.934 | 0.107 | 0.932 | 0.107 |
| | | 15 | 0.927 | 0.087 | 0.936 | 0.087 | 0.932 | 0.087 |
| | | 20 | 0.926 | 0.075 | 0.935 | 0.075 | 0.930 | 0.075 |
| **Setting 2** | ADE | 5 | 0.942 | 0.152 | 0.943 | 0.152 | 0.942 | 0.152 |
| | | 10 | 0.945 | 0.108 | 0.941 | 0.108 | 0.943 | 0.108 |
| | | 15 | 0.946 | 0.088 | 0.947 | 0.088 | 0.946 | 0.088 |
| | | 20 | 0.943 | 0.076 | 0.943 | 0.076 | 0.943 | 0.076 |
| | ODE | 5 | 0.931 | 0.152 | 0.936 | 0.152 | 0.933 | 0.152 |
| | | 10 | 0.927 | 0.107 | 0.932 | 0.108 | 0.929 | 0.107 |
| | | 15 | 0.925 | 0.088 | 0.934 | 0.088 | 0.930 | 0.088 |
| | | 20 | 0.919 | 0.076 | 0.930 | 0.076 | 0.925 | 0.076 |

Table S.16: Empirical average coverage probability (ACP) and average lengths (AL) of confidence intervals for ADE and ODE at the confidence level 95% under **Scenario 1** with $p = 200$.

| | | | $\mathcal{G}_1$ | | $\mathcal{G}_2$ | | $\mathcal{G}_3$ | |
| --- | --- | --- | --- | --- | --- | --- | --- | --- |
| | | $M$ | ACP | AL | ACP | AL | ACP | AL |
| **Setting 1** | ADE | 5 | 0.944 | 0.145 | 0.943 | 0.145 | 0.943 | 0.145 |
| | | 10 | 0.943 | 0.102 | 0.940 | 0.102 | 0.942 | 0.102 |
| | | 15 | 0.944 | 0.084 | 0.943 | 0.084 | 0.943 | 0.084 |
| | | 20 | 0.944 | 0.072 | 0.943 | 0.072 | 0.943 | 0.072 |
| | ODE | 5 | 0.929 | 0.144 | 0.929 | 0.144 | 0.929 | 0.144 |
| | | 10 | 0.924 | 0.102 | 0.923 | 0.102 | 0.923 | 0.102 |
| | | 15 | 0.922 | 0.083 | 0.926 | 0.083 | 0.924 | 0.083 |
| | | 20 | 0.921 | 0.072 | 0.926 | 0.072 | 0.923 | 0.072 |
| **Setting 2** | ADE | 5 | 0.945 | 0.146 | 0.942 | 0.146 | 0.943 | 0.146 |
| | | 10 | 0.943 | 0.103 | 0.940 | 0.103 | 0.942 | 0.103 |
| | | 15 | 0.943 | 0.084 | 0.940 | 0.084 | 0.941 | 0.084 |
| | | 20 | 0.942 | 0.073 | 0.942 | 0.073 | 0.942 | 0.073 |
| | ODE | 5 | 0.929 | 0.145 | 0.929 | 0.145 | 0.929 | 0.145 |
| | | 10 | 0.925 | 0.103 | 0.925 | 0.103 | 0.925 | 0.103 |
| | | 15 | 0.924 | 0.084 | 0.923 | 0.084 | 0.923 | 0.084 |
| | | 20 | 0.920 | 0.073 | 0.922 | 0.073 | 0.921 | 0.073 |

Table S.17: Empirical average coverage probability (ACP) and average lengths (AL) of confidence intervals for ADE and ODE at the confidence level 95% under **Scenario 2** with $p = 200$.

|  |  | $M$ | $\mathcal{G}_1$ ACP | $\mathcal{G}_1$ AL | $\mathcal{G}_2$ ACP | $\mathcal{G}_2$ AL | $\mathcal{G}_3$ ACP | $\mathcal{G}_3$ AL |
|---|---|---|---|---|---|---|---|---|
| **Setting 1** | ADE | 5 | 0.945 | 0.154 | 0.942 | 0.154 | 0.944 | 0.154 |
|  |  | 10 | 0.946 | 0.109 | 0.943 | 0.109 | 0.945 | 0.109 |
|  |  | 15 | 0.944 | 0.089 | 0.943 | 0.089 | 0.944 | 0.089 |
|  |  | 20 | 0.944 | 0.077 | 0.944 | 0.077 | 0.944 | 0.077 |
|  | ODE | 5 | 0.931 | 0.153 | 0.928 | 0.153 | 0.929 | 0.153 |
|  |  | 10 | 0.928 | 0.108 | 0.928 | 0.108 | 0.928 | 0.108 |
|  |  | 15 | 0.925 | 0.088 | 0.925 | 0.088 | 0.925 | 0.088 |
|  |  | 20 | 0.924 | 0.076 | 0.925 | 0.076 | 0.924 | 0.076 |
| **Setting 2** | ADE | 5 | 0.945 | 0.155 | 0.944 | 0.155 | 0.944 | 0.155 |
|  |  | 10 | 0.944 | 0.109 | 0.944 | 0.109 | 0.944 | 0.109 |
|  |  | 15 | 0.945 | 0.089 | 0.943 | 0.089 | 0.944 | 0.089 |
|  |  | 20 | 0.945 | 0.077 | 0.943 | 0.077 | 0.944 | 0.077 |
|  | ODE | 5 | 0.930 | 0.154 | 0.928 | 0.154 | 0.929 | 0.154 |
|  |  | 10 | 0.927 | 0.109 | 0.929 | 0.109 | 0.928 | 0.109 |
|  |  | 15 | 0.921 | 0.089 | 0.926 | 0.089 | 0.924 | 0.089 |
|  |  | 20 | 0.920 | 0.077 | 0.926 | 0.077 | 0.923 | 0.077 |

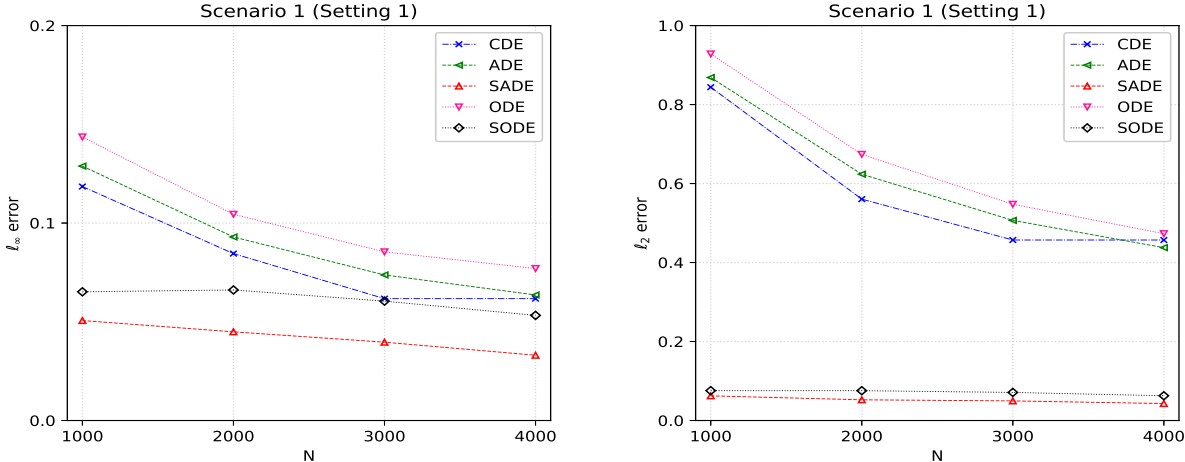

Figure S.1: The plot of $\ell_\infty$ and $\ell_2$ errors for $N \in \{1000, 2000, 3000, 4000\}$ under **Scenario 1** and **Setting 1**. The dashdot line with cross points, dashed line with left triangle points, dashed line with up triangle points, dotted line with down triangle points, and dotted line with diamond points respectively represent the results corresponding to CDE, ADE, SADE, ODE, and SODE.

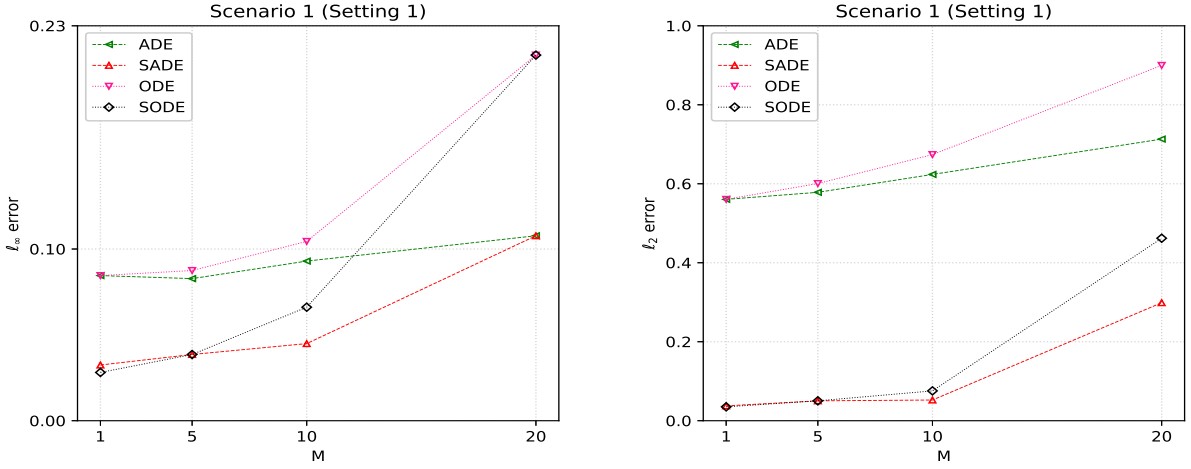

Figure S.2: The plot of $\ell_\infty$ and $\ell_2$ errors for $M \in \{1, 5, 10, 20\}$ under **Scenario 1** and **Setting 1** ($M = 1$ corresponds to the CDE). The dashed line with left triangle points, dashed line with up triangle points, dotted line with down triangle points, and dotted line with diamond points respectively represent the results corresponding to ADE, SADE, ODE, and SODE.

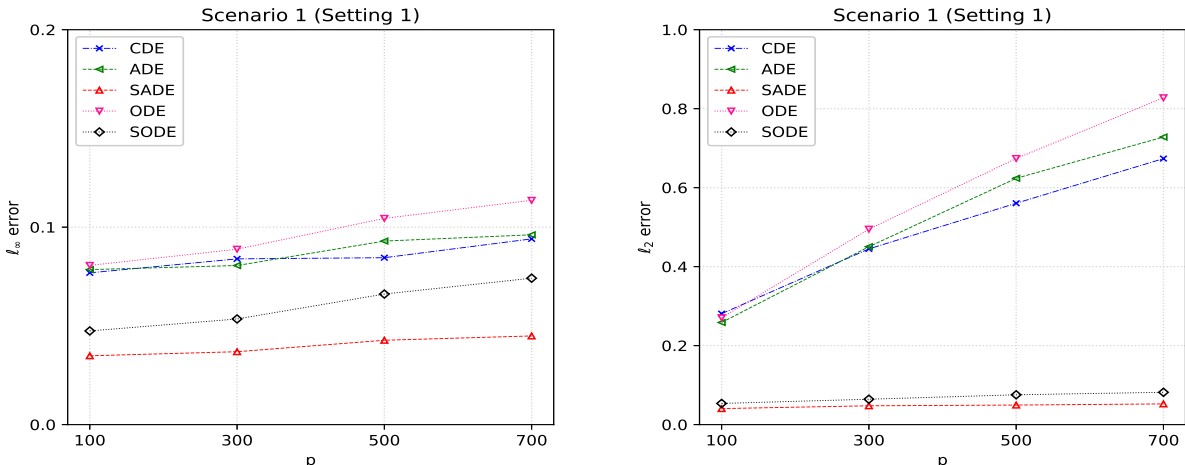

Figure S.3: The plot of $\ell_\infty$ and $\ell_2$ errors for $p \in \{100, 300, 500, 700\}$ under **Scenario 1** and **Setting 1**. The dashdot line with cross points, dashed line with left triangle points, dashed line with up triangle points, dotted line with down triangle points, and dotted line with diamond points respectively represent the results corresponding to CDE, ADE, SADE, ODE, and SODE.