# OpenReview forum: "Distributed inference for high-dimensional convoluted rank regression"
_SLADS/Section_B — Accepted by SLADS_Section_B_

### Review · Reviewer_B2U2 · 2026-02-08

**Summary Of Contributions:**

This paper studies distributed inference for high-dimensional convoluted rank regression (CRR). The authors extend CRR from the centralized setting to distributed environments and propose two debiased estimators: an averaging debiased estimator (ADE), which is communication-efficient, and a one-step debiased estimator (ODE), which is computation-efficient. For both estimators, the paper establishes asymptotic representations, high-dimensional Gaussian approximation results, and $\ell_\infty$ error bounds. In addition, sparse thresholded versions are introduced to improve $\ell_1$ and $\ell_2$ estimation accuracy. The theoretical findings are supported by simulation studies and a real data application.

**Audience:**

Yes

**Broader Impact Concerns:**

The paper presents a theoretical methodological contribution in distributed statistical inference. Any downstream risks would depend on specific applications rather than the proposed methods themselves.

**Claims And Evidence:**

Yes

**Requested Changes:**

1. The current theoretical development assumes that the data across machines are i.i.d. and share the same distribution. It would be valuable for the authors to clarify whether the proposed procedures can be extended to settings with covariate shift, where the distribution of $X$ may vary across sites. In practice, heterogeneous feature distributions are common in distributed systems. Since the construction of $J_h$ and its inverse relies on a common covariance structure, it is unclear whether the debiasing step remains valid under heterogeneous covariance matrices $\Sigma_k$ across machines.

2.  While many theoretical results are derived, the paper would benefit from additional interpretation and comparison, especially comparing with existing method about under what regimes on $M$, $n$ do the results match centralized estimators.

3. Since sparse thresholded estimators are proposed, it would be valuable to investigate whether model selection consistency or support recovery guarantees can be established. If not feasible, at leas a simulation should be add to see the selection performance compare to existing centralized methods

**Strengths And Weaknesses:**

Strengths:
1. The paper is generally well written and easy to follow.
2. The authors provide comprehensive theoretical analysis.
3. The distinction between ADE (communication-efficient) and ODE (computation-efficient) is conceptually clean and practically meaningful.

Weaknesses:
1. The analysis is conducted under a relatively standard i.i.d. distributed linear model.
2. Although many theoretical results are derived, the paper provides limited discussion on: how these results compare with existing distributed inference literature, whether the rates are optimal.
3. The paper introduces thresholded sparse versions (SADE and SODE), but no theoretical guarantees on support recovery or variable selection consistency are provided.

---

> ### Author Response · Authors · 2026-04-02
> **Response to Requested Changes and weakness 1 and 2**
>
> ## Response to Reviewer B2U2
>
> ### **Q1 The current theoretical development assumes that the data across machines are i.i.d. and share the same distribution. It would be valuable for the authors to clarify whether the proposed procedures can be extended to settings with covariate shift, where the distribution of $\boldsymbol X$ may vary across sites. In practice, heterogeneous feature distributions are common in distributed systems. Since the construction of $\boldsymbol{J}_h$ and its inverse relies on a common covariance structure, it is unclear whether the debiasing step remains valid under heterogeneous covariance matrices $\boldsymbol\Sigma_k$.**
>
> We thank the reviewer for the positive comments on the writing, the comprehensive theoretical development, and the conceptual distinction between ADE and ODE. We are encouraged that the reviewer found the paper easy to follow and viewed the communication-efficient/computation-efficient contrast as meaningful in practice.
>
> Regarding the concern that the original analysis was developed under a relatively standard i.i.d. distributed linear model, we agree that this point should be clarified more carefully. In the revised manuscript, we now separate the roles of ADE and ODE much more explicitly. For ODE, some degree of cross-machine homogeneity is indeed essential, because its computational advantage comes precisely from using only a single inverse Hessian proxy. This simplification is valid only when the local Hessian structures are identical or sufficiently close.
>
> By contrast, such a restriction is not intrinsic for ADE. Since ADE already computes a local debiasing matrix on each machine, in the revised manuscript we add a new discussion of a heterogeneity-aware ADE (HADE), which allows machine-specific covariance matrices $\Sigma_k$ and corresponding Hessians $J_{k,h}$. We further establish the asymptotic representation and Gaussian approximation for this heterogeneous extension. Therefore, after revision, the common-covariance assumption is no longer needed for ADE, while for ODE it remains the price paid for its major computational reduction. We believe this revision makes the scope of the two procedures much clearer and substantially improves the practical relevance of the manuscript.
>
> ### **Q2 While many theoretical results are derived, the paper would benefit from additional interpretation and comparison, especially comparing with existing method about under what regimes on $M, n$ do the results match centralized estimators.**
>
>
> We also appreciate the reviewer’s comment that the original manuscript did not sufficiently explain how our results compare with the existing distributed inference literature, or under what regimes our distributed estimators match the centralized estimator. In response, we have added a dedicated discussion subsection in the revised manuscript to address these issues directly
>
> More specifically, we now compare our results with several representative methods for distributed high-dimensional linear models, including one-shot debiasing, divide-and-conquer inference, surrogate-likelihood methods, and iterative sparse distributed learning. In this new discussion, we explain that, under the stated conditions of our main theorems, both ADE and ODE achieve the same final simultaneous-inference scale as the centralized debiased estimator, namely $
> O_p\left(\sqrt{\log p / N}\right)
> $ in the $\ell_\infty$ norm. Thus, in the asymptotic sense, our estimators match the centralized rate.
>
> At the same time, we now explicitly state that our nonasymptotic remainder conditions are somewhat more conservative than those in many least-squares-based distributed debiasing papers. The main reason is structural: the leading term in our debiasing analysis is a high-dimensional $U$-statistic induced by the pairwise convoluted rank loss, rather than a sum of independent score vectors. As a result, our proof relies on Gaussian approximation theory for high-dimensional $U$-statistics, which is technically more delicate and leads, in our current derivation, to more conservative logarithmic factors and machine-growth conditions in finite samples. We do not claim sharper nonasymptotic constants than least-squares-based methods, and we now make this point explicit in the revised manuscript.

---

> ### Author Response · Authors · 2026-04-02
> **Response to Requested Changes and weakness 3**
>
> ## Response to Reviewer B2U2
> ### **Q3 Since sparse thresholded estimators are proposed, it would be valuable to investigate whether model selection consistency or support recovery guarantees can be established. If not feasible, at leas a simulation should be add to see the selection performance compare to existing centralized methods**
>
> We also agree with the reviewer that, once sparse thresholded estimators are proposed, it is natural and important to ask whether they enjoy support recovery or model selection consistency guarantees. In the original submission, SADE and SODE were mainly justified through their improved $\ell_1$ and $\ell_2$ error bounds. In the revised version, we have strengthened this part substantially.
>
> Specifically, in the appendix we add new theory for the support of $\beta^*$. We now prove that when the threshold is chosen at the order $\sqrt{\log p / N}$, the sparse thresholded estimators satisfy a no-false-positives property with probability tending to one. Moreover, under a standard beta-min condition on the nonzero coefficients, we further establish exact support recovery, equivalently model selection consistency. We also provide the complete proof in the appendix. Therefore, after revision, the sparse versions are supported not only by improved $\ell_1$ and $\ell_2$ estimation error bounds, but also by explicit theoretical guarantees for variable selection.

---

### Review · Reviewer_4SAx · 2026-03-19

**Summary Of Contributions:**

This paper studies distributed statistical inference for high-dimensional convoluted rank regression (CRR), motivated by robust regression under outliers and heavy-tailed noise in large-scale distributed settings. The main methodological contributions are two debiased estimators: an averaging debiased estimator (ADE), designed to be communication-efficient, and a one-step debiased estimator (ODE), designed to be computation-efficient. The paper also introduces thresholded sparse variants (SADE/SODE), and provides asymptotic representations, Gaussian approximation results, confidence intervals, and error bounds. Empirically, the paper includes synthetic experiments and a GTEx gene-expression case study.

**Audience:**

Yes

**Broader Impact Concerns:**

Not applicable.

**Claims And Evidence:**

Yes

**Requested Changes:**

1. Report simulations under different $h$ values to verify insensitivity in the distributed setting. It would also be valuable to discuss whether the theoretical conditions on h interact with the number of machines M, though not required.
2. Comparisons with existing methods should be included in both the simulation study and the real data analysis.
3. Algorithmic descriptions (i.e., pseudocode) of the proposed methods should be provided to make the procedures easier to implement and reproduce.

**Strengths And Weaknesses:**

Strengths:
1. Well motivated problem: The problem is well motivated. Robust high-dimensional inference under distributed computation is important, and extending CRR to this setting is a natural and meaningful direction. The ADE/ODE split is conceptually clean: ADE targets low communication, whereas ODE targets low computation by estimating the inverse Hessian surrogate on only one machine.
2. Strong theoretical results: The paper offers a clear tradeoff: ADE requires $M$ separate Hessian inverse estimates but only one round of communication, while ODE requires only one Hessian inverse but aggregates gradients from all machines. The theoretical results show both achieve the same asymptotic representation and $\ell_\infty$ rate of $O_p(\sqrt{\log p / N})$, which matches the centralized oracle rate. The distinction between communication efficiency and computation efficiency is well-articulated and practically useful.
3. Comprehensive experiments: The simulation study is thorough, varying total sample size N, number of machines M, dimension p, error distributions, covariance structures, coefficient vectors, and index sets for confidence interval evaluation. Coverage probabilities consistently stay near the nominal 95% across all configurations, even under Cauchy errors where no moments exist. The computational time comparison concretely demonstrates the practical advantage. The supplementary material is also commendably extensive, with results across six error distributions and alternative threshold selection rules.

Weaknesses:
1. Unclear role of bandwidth h:
Compared with Zhou et al. (2024), the paper does not adequately discuss the impact of the bandwidth parameter h. The present paper appears to fix h=1 throughout without justification or any sensitivity analysis. This makes it difficult to assess the robustness of the proposed method to the choice of bandwidth.

2. Limited empirical comparisons:
The empirical evaluation only compares variants of the authors’ own proposed methods. Without including external benchmark methods, it is hard to understand the practical advantages and limitations of the CRR-based approach or the settings in which it is preferable. It would strengthen the paper if the authors could include at least one or two alternative robust distributed methods in both the simulation studies and the real-data analysis, and report confidence intervals and coverage alongside their current metrics. Releasing the code would also substantially improve the paper’s reproducibility and impact.

---

### Decision · Action_Editor_sPa3 · 2026-04-22

**Recommendation:** Accept as is

**Audience:**

Yes.

**Claims And Evidence:**

Yes.